# Improved RRT-Connect Algorithm Based on Triangular Inequality for Robot Path Planning

**DOI:** 10.3390/s21020333

**Published:** 2021-01-06

**Authors:** Jin-Gu Kang, Dong-Woo Lim, Yong-Sik Choi, Woo-Jin Jang, Jin-Woo Jung

**Affiliations:** 1Department of Computer Science and Engineering, Dongguk University, Seoul 04620, Korea; kanggu12@dongguk.edu (J.-G.K.); aehddn@gmail.com (D.-W.L.); skwndbth159@dongguk.edu (W.-J.J.); 2Department of Artificial Intelligence, Dongguk University, Seoul 04620, Korea; sik2230@dongguk.edu

**Keywords:** RRT-Connect, triangular inequality, rewiring, optimality, robot path planning

## Abstract

This paper proposed a triangular inequality-based rewiring method for the rapidly exploring random tree (RRT)-Connect robot path-planning algorithm that guarantees the planning time compared to the RRT algorithm, to bring it closer to the optimum. To check the proposed algorithm’s performance, this paper compared the RRT and RRT-Connect algorithms in various environments through simulation. From these experimental results, the proposed algorithm shows both quicker planning time and shorter path length than the RRT algorithm and shorter path length than the RRT-Connect algorithm with a similar number of samples and planning time.

## 1. Introduction

With the recent Fourth Industrial Revolution, interest in mobile robots has increased in various fields such as robotics, smart factories, and autonomous driving [1]. Classical mobile robot path-planning algorithms can be classified into three broad categories [2]. The first is the road map approach algorithm [3], which is easy to implement by designing a map that represents a path that can be moved and plan through it. The second is cell decomposition algorithm [4], which creates a path by dividing the configuration space into cells and connecting each cell using a graph. The last is the artificial potential field algorithm [5], which creates an artificial potential field and directs the robot to the goal according to the flow of potential power.

‘Optimality’ means always ensuring the optimal path. ‘Clearance’ indicates a lower probability of collision between obstacles and the robot. ‘Completeness’ means that if a path exists, it can always be found. Optimality, clearance, and completeness are considered important in these classical algorithms and have been the main focus of study [6]. Particularly if completeness is not guaranteed by the robot path-planning algorithm, there is a problem that the path may not be found in finite time. This is a fatal problem in robot path planning.

Recently, sampling-based path-planning algorithms [7,8,9,10,11,12] such as rapidly exploring random tree (RRT) [13], which is quicker and less computationally intensive than classical algorithms, have been attracting attention. The main purpose of sampling-based algorithms is to find a path that can reach the goal as quickly as possible using randomly extracted sample points (random sampling). Unlike classical algorithms, sampling-based algorithms have difficulty fully reflecting the optimality and completeness. Therefore, most sampling-based algorithms claim ‘Probabilistic completeness’, which explains that they can be probabilistically close to complete when random sampling is repeated infinitely [14]. This means that it is difficult to guarantee the ‘Planning time’ (first path finding time), which refers to how quickly the path can be planned from the start point to the goal point, and the ‘Convergence rate’, which means iterative sampling to bring the path closer to the optimum after the first path has been found. If the situation does not allow enough time to plan the path, it can create a path that is more different from the optimal path. Even so, the sampling-based algorithm is mainly used in dynamic environments because it enables quicker path planning with very little planning time compared to classical algorithms.

To overcome these limitations of planning time and convergence rate, many studies are being conducted to expand the RRT algorithm. The RRT-Connect [15] algorithm finds a connected path more quickly than the RRT algorithm by setting the start point and goal point as the roots of separate trees and expanding both trees alternately. In addition, there are algorithms that optimize paths based on the principle of triangular inequality, such as RRT*-Smart algorithm [16] and Quick-RRT* algorithm [17], to derive a path that is close to the optimal. Many algorithms [18,19,20,21] that extend the RRT algorithm have been studied.

The above algorithms show more efficient performance by improving the RRT algorithm to overcome the limitations of sampling-based methods but they are still not perfect. Their limitations include being unable to derive the optimal length and there is room for improvement in terms of the number of operations and time. For example, the RRT* algorithm has rewiring(search for the parent node as a via point nearby a newly inserted node, where the addition of path length from the start point to the via point and path length from the via point to the newly inserted node in the tree is the optimized, and change the neighboring nodes to optimize the path length) and neighbor search (search for nodes nearby the node to be newly inserted in the tree) processes to obtain shorter path lengths than the RRT algorithm [18]. However, there is an efficiency trade-off in this process. In other words, while the convergence rate has improved, the planning time has significantly increased [22]. Therefore, the RRT* algorithm cannot be said to be better than the RRT algorithm in all performance metrics and it can be said that the RRT* algorithm gets closer to the optimum at the expense of planning time.

To overcome the limitation of getting closer to the optimum at the expense of planning time, this paper proposes a triangular inequality-based RRT-Connect algorithm that finds an ancestor node as a via point, where the addition of path length from the start point to the via point and path length from the via point to the newly inserted node is the most optimized, based on the principle of triangular inequality and RRT-Connect. The proposed algorithm shortens the planning time while also pursuing optimization through rewiring. In addition, we will verify the efficiency by comparing the RRT and RRT-Connect algorithms from previous studies through simulation experiments. As a result, this paper shows that the proposed algorithm has a shorter path length than the RRT and RRT-Connect algorithms without sacrificing other performance measures such as the number of samples or planning time.

The scope of the research we will cover is how much more quickly it can find the path and how much shorter the path is. This is because in a dynamic environment, it is more important to find a navigable path. In a dynamic environment, there may not be enough time for convergence. In other words, the purpose of our proposed algorithm is to improve the RRT-Connect algorithm so that it can find a shorter path over the same planning time (computation time before convergence or computation time for first path finding).

Figure 1 shows an overview of the three main algorithms covered in this paper: RRT, RRT-Connect, and the proposed algorithm. In this figure, the start *q_start_* and goal points are *q_goal_*, respectively. The RRT algorithm in Figure 1a shows that the path is expanded in a tree structure and the RRT-Connect algorithm in Figure 1b shows that the trees that are expanded at the start and goal points attract and connect each other. The proposed algorithm in Figure 1c shows that the RRT-Connect algorithm was rewired into a triangular inequality during path planning.

In this paper, Section 2 introduces the RRT algorithm, Section 3 introduces the RRT-Connect algorithm, and the triangular inequality-based RRT-Connect algorithm is proposed in Section 4. In detail, Section 4.1 shows the pseudocode of the proposed rewiring method through the principle of triangular inequality, which can be applied to the RRT-Connect algorithm, Section 4.2 shows the mathematical modeling of the proposed algorithm, Section 4.3 and Section 4.4 show the pseudocode of each method of the RRT-Connect algorithm applying the proposed rewiring method, and Section 4.5 shows the path-planning process for the proposed algorithm that applies the proposed rewire method to the RRT-Connect algorithm. Section 5 shows the experimental environment and results to check the performance of the proposed algorithm and Section 6 presents the conclusion.

## 2. Rapidly Exploring Random Tree (RRT) Algorithm

Rapidly exploring Random Tree (RRT) algorithm [13] is the most representative sampling-based path-planning algorithm. The RRT algorithm plans a path by gradually expanding a tree with a root node at the start point using random sampling. It is designed to handle non-holonomic constraints and high degrees of freedom [12].

When a random sample is generated in the configuration space, it tries to connect at a point separated by a preset step length from the node nearest to the random sample among nodes constituting the tree with the step length. If tree connections are possible, nodes are added to create an extended tree.

As mentioned in the introduction, this sampling-based path-planning algorithm uses randomly generated sample points to find a path that can reach the goal as quickly as possible, so it is difficult to sufficiently reflect the optimality and completeness.

Figure 2 shows the path-planning process of the RRT algorithm. Figure 2a shows that *q_new_* is created at the node position *q_near_* of the tree *T* nearest to the random sample position *q_rand_*. Figure 2b shows the resultant path *R* among several candidate paths to the start position *q_start_* and the goal position *q_goal_*.

## 3. RRT-Connect Algorithm

Path planning through the RRT algorithm may have a disadvantage in that since random samples appear with the same probability in all regions, the tree easily extends even in a direction irrespective of the goal, resulting in a long planning time and inefficiency. The RRT-Connect algorithm [15] proposed later has two new ideas as the method to compensate for the disadvantage of the RRT algorithm.

The first is that the start and goal points are each inserted as root nodes and extended in each direction alternately. The two trees extending from the start point and the goal point expand as if attracting one another (which prevents the tree and is a disadvantage of the RRT algorithm) is in a direction irrespective of the goal. This enhances the disadvantage of the planning time required to search for a path. The second is the concept of ‘Extend’, which continues extending to the other side of the tree if there are no collisions with obstacles when the tree extends. Through this, unlike the RRT algorithm that extends the maximum extension length when the sample is generated and is inserted into the tree, the tree continues to expand in the direction of the goal if there is no collision with an obstacle, so the path can be planned more quickly.

Path planning through the RRT-Connect algorithm can find a path quicker than the RRT algorithm, but the ‘Extend’ method does not work properly in complex environments with narrow paths and many obstacles and it can be difficult. In addition, the path planned using the RRT-Connect algorithm is far from the optimal length, so it does not properly reflect optimality.

### 3.1. Pseudocode of the RRT-Connect Algorithm

This section shows the pseudocode of the RRT-Connect algorithm used in the experiment in this paper that was designed based on [15] in which the RRT-Connect algorithm was proposed. The RRT-Connect algorithm can be represented by a main algorithm (A1) and two main methods (A2 and 3).

Algorithm 1 shows the pseudocode of RRT-Connect algorithm. Both of the two initial trees *T_a_* and *T_b_* have *q_start_* and *q_goal_* as root nodes and these two trees randomly sample *N* times and aim to reach each other during their expansion. Unlike RRT, the RRT-Connect algorithm is divided into two methods: ‘Extend’ and ‘Connect’. The ‘Extend’ method (A2) creates *q_new_* from *q_rand_* in *T_a_* and extends from *T_b_* to the *q_new_* direction of *T_a_*, and the ‘Connect’ method (A3) determines whether the two trees *T_a_* and *T_b_* have reached each other; if they do, merge them into one tree to obtain a path *P_reach_* between the root nodes *q_start_* and *q_goal_* of the two trees.

**Algorithm 1** Pseudocode of the RRT-Connect Algorithm**Input:***q_start_* ← Start Point Position*q_goal_* ← Goal Point Position*λ* ← Step Length*C* ← Position Set of All Boundary Points in All Obstacles*Ν* ← Number of Random Samples**Output:***R* ← Result of Path *R***Initialize:***T_a_* ← ***Null*** Tree*T_b_* ← ***Null*** Tree*d_shorter_* ← 0
**Begin**
*RRT-Connect*
**Procedure**
1*T_a_* ← **Insert** Root Node<*q_start_*> to *T_a_*2*T_b_* ← **Insert** Root Node<*q_goal_*> to *T_b_*3**While** 1 ← *n* to *N*
**do**4 **Generate** *n*-th Random Sample5 *q_rand_* ← Position of *n*-th Random Sample6 **If Not**
*Extend*(*T_a_, T_b_, q_newB_ ← **Null**, q_rand_, λ, C*) **then**7  **If** *Connect*(*P_reach_* ← ***Null*** Path, *T_a_*, *T_b_*, *q_newB_*, *λ*) **then**8   *d_reach_* ← Distance of *P_reach_*9   **If**
*d_shorter_*
**=** 0 **or**
*d_shorter_**>** d_reach_*
**then**10    *R ← P_reach_*11    *d_shorter_ ← d_reach_*12 *Swap*(*T_a_, T_b_*)
**End**
*RRT-Connect*
**Procedure**


When a path is created by the ‘Connect’ method, the distance *d_reach_* is calculated for the path *P_reach_* from *q_start_* to *q_goal_*. At this time, if *d_reach_* is smaller than *d_shorter_*(the shortest path length until now) or *P_reach_* is the first path found (i.e., *d_shorter_* = 0), the resultant path R becomes *P_reach_*, and *d_shorter_* becomes *d_reach_*. At the end of the next *N* sampling, *R* becomes the final planned path. If the number of random sampling remains, the above process is repeated.

### 3.2. Pseudocode of the Extend Method from the RRT-Connect Algorithm

This section introduces the ‘Extend’ method used in pseudocode (A1) of the RRT-Connect algorithm in Section 3.1.

Algorithm 2 shows the pseudocode of the ‘Extend’ method in the RRT-Connect algorithm. The *isInside* function determines whether *q_rand_* is inside a circle (or *n*-sphere) with the node position *q_near_* of the tree *T_a_* nearest the *q_rand_* position as the center and *λ* as the radius. If it is not located inside (*False*), *q_newA_* becomes the intersection of the circle (or *n*-sphere) with *q_near_* as the center and *λ* as the radius, and the line segment connecting *q_rand_* and *q_near_*. If it is determined that there is no obstacle between *q_newA_* and *q_near_* by the *isTrapped* function (*False*), *q_newA_* is inserted into the tree as a child node of *q_near_* of *T_a_*. If there is an obstacle (*True*), the ‘Extend’ method returns *True* (*f_trap_*) and terminates. Otherwise, it proceeds with the remaining process and returns *False* (*f_trap_*) when the process ends.
**Algorithm 2** Pseudocode of the original ‘Extend’ method from the RRT-Connect Algorithm**Input:***T_a_* ← Tree *T_a_* from *RRT-Connect**T_b_* ← Tree *T_b_* from *RRT-Connect**q_newB_* ← Position *q_newB_* from *RRT-Connect**q_rand_* ← Position *q_rand_* from *RRT-Connect**λ* ← Step Length *λ* from *RRT-Connect**C* ← Position Set *C* from *RRT-Connect***Output:***f_trap_* ← Result of Boolean *f_trap_**T_a_* ← Result of Tree *T_a_* /**/Return by Reference***T_b_* ← Result of Tree *T_b_* /**/Return by Reference***q_newB_* ← Result of Position *q_newB_* /**/Return by Reference****Initialize:***f_trap_* ← ***False*****Begin ***Extend***Procedure from***RRT-Connect*1*q_near_* ← **Find** Position of Nearest Node in *T_a_* from *q_rand_*2**If Not***isInside*(*q_near_, q_rand_, λ*) **then**3 *q_newA_* ← Position of the Intersection Point between the Line Segment connecting *q_rand_* and *q_near_* and a Circle with Radius *λ* centered at *q_near_* /**/2D:** Circle**, 3D:** Sphere**, …**4**Else**5 *q_newA_* ← *q_rand_*6**If***isTrapped*(*q_newA_, q_near_, C*) **then**7 *f_trap_* ← ***True***8**Else**9 *T_a_* ← **Insert** Node<*q_newA_**>* and Edge<*q_newA_*, *q_near_*> to *T_a_*10 *q_near_* ← **Find** Position of Nearest Node in *T_b_* from *q_newA_*11 **If**
*isInside*(*q_near_, q_newA_, λ*) **then**12  *q_newB_* ← *q_near_*13 **Else**14 *q_newB_* ← Position of Intersection Point between Line Segment connecting *q_newA_* and *q_near_*, and Circle with Radius *λ* centered at *q_near_* /**/2D:** Circle**, 3D:** Sphere**, …**15  **While Not**
*isTrapped*(*q_newB_, q_near_, C*) **do**16   *T_b_* ← **Insert** Node<*q_newB_*> and Edge<*q_newB_*, *q_near_*> to *T_b_*17   **If Not**
*isInside*(*q_newA_, q_newB_, λ*) **then**18    *q_near_* ← *q_newB_*19    *q_newB_* ← Position of Intersection Point between Line Segment connecting *q_newA_* and *q_near_*, and Circle with Radius *λ* centered at *q_near_* /**/2D:** Circle**, 3D:** Sphere**, …**20   **Else**21    ***Break*****End***Extend***Procedure from***RRT-Connect*

This is the process of making *T_a_* and *T_b_* reach each other: First, the node *T_b_* nearest to *q_newA_* becomes the new *q_near_*. At this time, using the *isInside* function, it is determined whether *q_newA_* is inside a circle (or *n*-sphere) with *q_near_* as the center and *λ* as the radius, and if it is located inside (*True*), *q_newB_* becomes *q_near_* and is located inside. If not (*False*), *q_newB_* becomes the intersection of the circle (or *n*-sphere) with *q_near_* as the center and *λ* as the radius and the line segment connecting *q_newA_* and *q_near_*. If *q_newB_* is created, then the following process is repeated until it can determine whether there is an obstacle between *q_newB_* and *q_near_* by the *isTrapped* function and if there is the obstacle between them (*True*) or if *q_newB_* reaches *q_newA_* by the *isInside* function.

If there is no obstacle between *q_newB_* and *q_near_* (*False*), insert *q_newB_* into *T_b_* as a child node of *q_near_*. At this time, if the *isInside* function determines that *q_newB_* has not reached the *λ* radius with *q_newA_* as the center (*False*), *q_near_* becomes *q_newB_* and a new *q_newB_* will created from this *q_near_*.

Figure 3 shows the ‘Extend’ method in the RRT-Connect algorithm. In detail, it shows that the first *q_newA_* is created, and *q_newB_* is created with radius of length *λ* in the direction of *q_newA_* from the *q_near_* position in the figure. Clearly, *T_b_* extends in the *T_a_* direction for reach.

### 3.3. Pseudocode of the Connect Method from the RRT-Connect Algorithm

This section introduces the ‘Connect’ method used in pseudocode (A1) of the RRT-Connect algorithm in Section 3.1.

Algorithm 3 shows the pseudocode of the ‘Connect’ method in the RRT-Connect algorithm. Here, *T_a_*, *T_b_*, and *q_newB_* are from the ‘Extend’ method (A2).
**Algorithm 3** Pseudocode of the Original ‘Connect’ Method from the RRT-Connect Algorithm**Input:***P_reach_* ← Path *P_reach_* from *RRT-Connect**T_a_* ← Tree *T_a_* from *RRT-Connect**T_b_* ← Tree *T_b_* from *RRT-Connect**q_newB_* ← Position *q_newB_* from *RRT-Connect**λ* ← Step Length *λ* from *RRT-Connect***Output:***f_reach_* ← Result of Boolean *f_reach_**P_reach_* ← Result of Path *P_merged_* //**Return by Reference****Initialize:***f_reach_* ← ***False*****Begin***Connect***Procedure from***RRT-Connect*1**If***isInside*(*q_newA_, q_newB_, λ*) **then**2 *P_a_* ← Path from Root Node [*q_start_*] to Last Inserted Node [*q_newA_*] in *T_a_*3 *P_b_* ← Path from *q_newB_* to Root Node [*q_goal_*] in *T_b_*4 *P_connect_* ← Path from Last Inserted Node [*q_newA_*] in *T_a_* to *q_newB_* in *T_b_*5 *P_merged_* ← **Merge Path**
*P**_a_* to *P_b_* via *P_connect_*6 *f_reach_* ← ***True*****End***Connect***Procedure from***RRT-Connect*

The tree merging process is as follows: create a path *P_a_* from the root node (*q_start_*) of *T_a_* to the last inserted node (*q_newA_*), and a path *P_b_* from *q_newB_* of *T_b_* to the root node (*q_goal_*). Then, create a path *P_connect_* from *q_newB_* of *P_b_* to the last inserted node (*q_newA_*) of *T_a_* and merge in the order of *P_a_*, *P_connect_*, and *P_b_*, thereby completing planning the path *P_merged_* from *q_start_* to *q_goal_*. After this, it returns *True* (*f_trap_*), and the ‘Connect’ method ends.

Figure 4 shows the ‘Connect’ method in the RRT-Connect algorithm. If the *q_newB_* of *T_b_* is extended in the direction of the *q_newA_* by the ‘Extend’ method shown in Figure 3, the point where the two trees merge (when *q_newB_* has expanded in the direction of *q_newA_* where *T_a_* enters the *λ* radius centered at *q_newA_*) with each other is the part marked as ‘Connect’. As a result, the path *P_a_* becomes from the position *q_start_* to the position *q_newA_* in *T_a_*, the path *P_connect_* goes from position *q_newA_* to position *q_newB_* and the path *P_b_* goes from position *q_newB_* to position *q_goal_* in *T_b_*. The merged path *P_merged_* goes from *q_start_* to *q_goal_*.

## 4. Proposed Triangular Inequality-Based RRT-Connect Algorithm

The proposed triangular inequality-based RRT-Connect algorithm is a rewire based on the principle of triangular inequality between nodes on a path planned in the RRT-Connect algorithm, so it is closer to the optimal compared to the RRT-Connect. This is like the RRT*-Smart algorithm [16] and Quick-RRT* [17] algorithms, which shorten their paths using the triangular inequality principle for the RRT algorithm. In this paper, the rewire part based on the triangular inequality principle is called the ‘Triangular-Rewiring’ method.

The proposed triangular inequality-based RRT-Connect algorithm requires the following assumptions:There is only one start point and one goal point even though the goal point may be changed incrementally as time goes on.The robot is capable of omnidirectional motion.

Therefore, this chapter introduces the proposed ‘Triangular-Rewiring’ method for the RRT-Connect algorithm, and performs mathematical modeling to confirm the validity that the proposed ‘Triangular-Rewiring’ method is always shorter when applied to the RRT-Connect algorithm. After checking through, we will propose how to apply the ‘Triangular-Rewiring’ method to the RRT-Connect algorithm.

The method of applying the RRT-Connect algorithm of the proposed ‘Triangular-Rewiring’ method is proposed when a new node is inserted into the tree in the ‘Extend’ method (A2) and ‘Connect’ method (A3), the main methods of the RRT-Connect algorithm introduced in Chapter 3. It is inserted after rewiring (or after determining) through the ‘Triangular-Rewiring’ method. That is, this chapter introduces the ‘Extend’ and ‘Connect’ methods to which the proposed ‘Triangular-Rewiring’ method is applied.

### 4.1. Pseudocode of the Proposed Triangular-Rewiring Method for the Improved RRT-Connect Algorithm

This section introduces the ‘Triangular-Rewiring’ method for the proposed triangular inequality-based RRT-Connect algorithm.

Algorithm 4 shows the pseudocode of the ‘Triangular-Rewiring’ method applicable in the ‘Extend’ (A2) and ‘Connect’ (A3) methods of the RRT-Connect algorithm. When inserting a new node and edge in *T_a_* or *T_b_* in the ‘Extend’ method (A5), when a tree *T_merged_* (*P_merged_*) in which *T_a_* and *T_b_* trees are merged in the ‘Connect’ method is created (A6), rewiring is performed on the tree *T*.
**Algorithm 4** Pseudocode of the Proposed ‘Triangular-Rewiring’ Method for the RRT-Connect Algorithm**Input:***q_child_* ← Position {*q_new_/q_newA_/q_newB_*} from {*Extend/Connect*}*q_parent_* ← Position *q_near_* from {*Extend/Connect*}*T* ← Tree {*T_merged_/T_a_/T_b_*} from {*Extend/Connect*}*C* ← Position Set *C* from {*Extend*/*Connect*}**Output:**{*T_merged_*/*T_a_*/*T_b_*} ← Result of *T***Begin***triangularRewiring***Procedure from***Extend, Connect*1*q_ancestor_* ← Position of Parent Node of *q_parent_* in *T*2**If Not***isTrapped*(*q_ancestor_, q_child_, C*) **then**3 *T* ← **Delete** Node<*q_parent_**>*, Edge<*q_parent_*, *q_child_*> and Edge<*q_parent_*, *q_ancestor_*> from *T*4 *q_parent_* ← *q_ancestor_*5 *q_ancestor_* ← Position of Parent Node of *q_ancestor_* in *T*6 **While Not**
*q_ancestor_* = ***Null***
**do**7  **If Not**
*isTrapped*(*q_ancestor_, q_child_, C*) **then**8   *T* ← **Delete** Node<*q_parent_**>* and Edge<*q_parent_*, *q_ancestor_*> from *T*9   *q_parent_* ← *q_ancestor_*10   *q_ancestor_* ← Position of Parent Node of *q_ancestor_* in *T*11  **Else**12   ***Break***13 *T* ← **Insert** Edge<*q_parent_*, *q_child_*> to *T*14**Else**15 *T* ← **Insert** Node<*q_child_**>* and Edge<*q_child_*, *q_parent_*> to *T***End***triangularRewiring***Procedure from***Extend, Connect*

In the ‘Extend’ and ‘Connect’ methods, *q_new_* (or *q_newA_* or *q_newB_*) is inserted as a *q_child_* and *q_near_* is inserted as a candidate for the node’s parent node. From *q_parent_*, the node’s parent node (a second ancestor node candidate based on *q_child_*) is called *q_ancestor_*. Next, it is determined whether an obstacle exists between *q_ancestor_* and *q_child_* (using the *isTrapped* function). If there is an obstacle (*True*), the ‘Triangular-Rewiring’ process is skipped and *q_child_* is inserted into the child node of *q_parent_* in *T* such that the contents of the ‘Extend’ and ‘Connect’ methods from the RRT-Connect algorithm are the same. If there is no obstacle (*False*), the ‘Triangular-Rewiring’ process proceeds.

The ‘Triangular-Rewiring’ process is as follows: Delete node where position *q_parent_* and the edges between *q_child_* and *q_ancestor_* nodes connected to *q_parent_*. In other words, it disconnects the existing *q_parent_* and *q_child_* and prepares to connect *q_child_* to *q_ancestor_*, the candidate parent node of *q_child_*. Again, *q_parent_* becomes its parent node *q_ancestor_* and *q_ancestor_* becomes the parent node of *q_ancestor_*. Then, as previously done, determine whether an obstacle exists between *q_ancestor_* and *q_child_* (using the *isTrapped* function). This iterative process continues until no *q_ancestor_* exists (when no parent node exists for the previous *q_ancestor_*, i.e., when *q_ancestor_* is *q_start_*) or an obstacle exists between *q_child_* and *q_ancestor_*. Then, in tree *T*, the last created *q_parent_* is inserted as the parent node of *q_child_*.

### 4.2. Mathematical Modeling of the Proposed Triangular Inequality-Based RRT-Connect Algorithm

This section introduces the mathematical modeling of the proposed triangular inequality-based RRT-Connect algorithm. The results show that the proposed algorithm is more efficient in terms of path length than the RRT-Connect algorithm. For reference, this mathematical modeling is based on a two-dimensional Euclidean space.

Equations (1) and (2) define the path length ${𝕕}_{n}\left(q_{i}\right)$ between an arbitrary node *q_i_* and its parent node in the RRT algorithm
(1)$$D\left(q_{i},\;\xi \left(q_{i}\right)\right)=\sqrt{{\left(\xi \left(q_{i}\right)\cdot x-q_{i}\cdot x\right)}^{2}+{\left(\xi \left(q_{i}\right)\cdot y-q_{i}\cdot y\right)}^{2}},$$
(2)$$∴{𝕕}_{n}\left(q_{i}\right)=D\left(\xi^{n}\left(q_{i}\right),\xi^{n+1}\left(q_{i}\right)\right)$$

Here, *q_i_* refers to the *i*-th inserted arbitrary node and takes the *x* and *y* coordinate values of the node as an element. The *ξ* function receives an arbitrary node as a variable and returns the parent node of this node. Equation (1) obtains the distance between an arbitrary node *q_i_* and its parent node, which can be summarized as a function ${𝕕}_{n}$ as in Equation (2). Here, *n* is the distance between the ancestor node and its parent node, based on an arbitrary node. That is, the *ξ* function to the power of *n* (*n* ≥ 0) can be represented as $\xi^{n}\left(q_{i}\right)∶=\overset{n}{\left(\overbrace{\xi \circ \xi \circ \ldots \circ \xi }\right)}\left(q_{i}\right)$; when *n* is 0, $\xi^{0}\left(q_{i}\right)∶=q_{i}$ holds.

In addition, consider starting with an arbitrary node *q_i_* and going back to the parent node to find the distance between the *n*-th ancestor node and the (*n* + 1)-th ancestor node; this can be represented as $D\left(\xi^{n}\left(q_{i}\right),\xi^{n+1}\left(q_{i}\right)\right)$.

Equations (3) and (4) show the path length $D_{R}$ from the start position *q_start_* to the goal position *q_goal_* by the RRT algorithm
(3)$$\xi^{\delta +1}\left(q_{goal}\right)=q_{start},$$
(4)$$∴D_{R}=\overset{\delta }{\underset{n=0}{\sum }}{𝕕}_{n}\left(q_{goal}\right)$$

Equation (3) shows when the (*δ* + 1)-th ancestor node from *q_goal_* is *q_start_*, where *δ* is the upper limit of $\overset{\delta }{\underset{n=0}{\sum }}{𝕕}_{n}\left(q_{goal}\right)$ for obtaining the path length $D_{R}$ in Equation (4). In other words, Equation (4) is the sum of the distances from *q_goal_* to the first ancestor node (parent node) of *q_goal_* and the distance from the first ancestor node (parent node) of *q_goal_* to the second ancestor of *q_goal_*, …, and (*δ* − 1)-th ancestor node to the *δ*-th ancestor node (*q_start_*).

Figure 5 shows an abstract process of the ‘Triangular-Rewiring’ method. As shown in Figure 5a, if the parent node of *q_child_* is *q_parent_*, the parent node of *q_parent_* is *q_ancestor_*, and *q_ancestor_* is the second ancestor of *q_child_*, this can be represented as Equation (5):(5)$$q_{ancestor}=\xi \left(q_{parent}\right)=\xi^{2}\left(q_{child}\right)$$

If the distances between the edges connecting each node are the *α* between *q_child_* and *q_parent_*, the *β* between *q_parent_* and *q_ancestor_*, and the *γ* between *q_child_* and *q_ancestor_* is as shown in Figure 5c, this can be represented as Equation (6) using the principle of the triangular inequality
(6)α+β≥ γ

Equations (7) and (8) show the distance relationship between the ancestor nodes of *q_child_*
(7)$$\left(q_{child},\;\xi \left(q_{child}\right)\right)=\alpha ,\;D\left(\xi \left(q_{child}\right),\;\xi^{2}\left(q_{child}\right)\right)=\beta ,\;D\left(q_{child},\;\xi^{2}\left(q_{child}\right)\right)=\gamma ,$$
(8)$$∴D\left(q_{child},\;\xi \left(q_{child}\right)\right)+D\left(\xi \left(q_{child}\right),\;\xi^{2}\left(q_{child}\right)\right)\geq D\left(q_{child},\;\xi^{2}\left(q_{child}\right)\right)$$

Equation (7) can be summarized as Equation (8) by substituting Equation (5), which represents the relationship between the *n*-th ancestor nodes of *q_child_*, with the distance as Equation (1) in Equation (6), which represents the distance between each node as a triangular inequality.

Equations (9)–(15) show that the path of the RRT algorithm applying the ‘Triangular-Rewiring’ method is always shorter or equal to that planned by the original RRT algorithm. Equation (9) shows the sequence index *k_j_* to compare the distance 𝕦 when applying the ‘Triangular-Rewiring’ method with distance 𝕕 when this method is not applied
(9)$$k_{j}=\tau_{j}+{k^{\prime }}_{j},\;{k^{\prime }}_{j}=\left\{\begin{array}{cc} 0, & j=0\\ k_{j-1}+1, & j\geq 1 \end{array}\right.$$

Here, *j* is a sequence index for 𝕦. That is, *k_j_* can be considered a sequence index for 𝕕. Currently, $\tau_{j}$ is the number of times that rewiring occurs in the *j*-th.

If this is summarized by Equation (1) for a distance based on an arbitrary node *q_i_*, it is as Equation (10). For example, as shown in Figure 5, if *j* is 0 and 1 a rewire occurs ($\tau_{0}$ = 1), it can be represented in combination with the distance relationship of Equation (8) for *q_child_*, as in Equation (11)
(10)$${𝕦}_{k_{j}}\left(q_{i}\right)=D\left(\xi^{{k^{\prime }}_{j}}\left(q_{i}\right),\;\xi^{k_{j}+1}\left(q_{i}\right)\right),$$
(11)$${𝕕}_{0}\left(q_{child}\right)+{𝕕}_{1}\left(q_{child}\right)=\overset{1}{\underset{n=0}{\sum }}{𝕕}_{n}\left(q_{child}\right)\geq {𝕦}_{k_{0}=1}\left(q_{child}\right)$$

The result of Equation (11) can be generalized as shown in Equation (12)
(12)$$∴\overset{k_{j}}{\underset{n=0}{\sum }}{𝕕}_{n}\left(q_{i}\right)\geq {𝕦}_{k_{j}}\left(q_{i}\right)$$

For 𝕕 based on an arbitrary node *q_i_*, the path length $\overset{k_{j}}{\underset{n=j}{\sum }}{𝕕}_{n}$ from the *j*-th to *k_j_*-th arbitrary sequence index is always longer or equal to the distance ${𝕦}_{k_{j}}$ of the *k_j_*-th sequence index. That is, in an arbitrary path, it can be confirmed that the distance 𝕦 rewired by the ‘Triangular-Rewiring’ method is at least equal (if the distances of 𝕕 and 𝕦 are the same, the rewired line segments are on a straight line) or always shorter than 𝕕 when not rewired.

Figure 6 shows the ‘Triangular-Rewiring’ process for the path from *q_start_* to *q_goal_* based on Equations (5)–(12) (at this time, it is assumed that the node of the path shown in the figure is not positioned in a straight line). As shown in Figure 6b, a total of two rewires occurred ($\tau_{0}=2$) between *q*_0_ and *q*_3_ ($\xi^{3}\left(q_{0}\right)$), and a total of one rewire occurred ($\tau_{3}=1$) between *q*_5_ ($\xi^{5}\left(q_{0}\right)$) and *q*_7_ ($\xi^{7}\left(q_{0}\right)$). In that case, as shown in Figure 6e, *k*_0_ is 2, *k*_1_ is 3, *k*_2_ is 4, and *k*_3_ is 6 according to Equation (9).

Comparing Figure 6c,e, according to Equation (7), the rewired distance ${𝕦}_{2}\left(q_{0}\right)$ is shorter than the path length $\overset{2}{\underset{n=0}{\sum }}{𝕕}_{n}\left(q_{0}\right)$ from ${𝕕}_{0}$ to ${𝕕}_{2}$ and the rewired distance ${𝕦}_{6}\left(q_{0}\right)$ is shorter than the path length $\overset{6}{\underset{n=5}{\sum }}{𝕕}_{n}\left(q_{0}\right)$ from ${𝕕}_{5}$ to ${𝕕}_{6}$. That is, when comparing before applying the ‘Triangular-Rewiring’ method in Figure 6a and after applied this method in Figure 6f, the path afterward looks shorter.

Equations (13) and (14) show the path length $D_{R}$ when the ‘Triangular-Rewiring’ method is not applied and the path length $U_{R}$ when the method has been applied for an arbitrary path (start position: *q_start_*, goal position: *q_goal_*), as shown in Figure 6
(13)$$k_{\varphi }=\delta ,$$
(14)$$D_{R}=\sum_{n=0}^{\delta }{𝕕}_{n}\left(q_{goal}\right)=\sum_{j=0}^{\varphi }\sum_{n={k^{\prime }}_{j}}^{k_{j}}{𝕕}_{n}\left(q_{goal}\right),\;U_{R}=\sum_{j=0}^{\varphi }{𝕦}_{k_{j}}\left(q_{goal}\right)$$

Equation (13) shows the upper limit when the index *n* of *d* is *δ* in Equation (3); when this is substituted into the sequence index *k_j_*, if *k_j_* is *δ*, *j* becomes *φ*. In that case, as in Equation (14), $D_{R}$ is used to compare the $\overset{\delta }{\underset{n=0}{\sum }}{𝕕}_{n}\left(q_{goal}\right)$ shown in Equation (4) with $U_{R}$, reflecting the sequence *k_j_*. It can be represented as $\overset{\varphi }{\underset{j=0}{\sum }}\overset{k_{j}}{\underset{n={k^{\prime }}_{j}}{\sum }}{𝕕}_{n}\left(q_{goal}\right)$, and $U_{R}$ can be represented as $\overset{\varphi }{\underset{j=0}{\sum }}{𝕦}_{k_{j}}\left(q_{goal}\right)$.

Equation (15) shows when the equation summarized in Equation (14) is substituted into Equation (12)
(15)$$∴D_{R}\geq U_{R}$$

Finally, as can be confirmed using Equation (15), $U_{R}$ as a result of applying the ‘Triangular-Rewiring’ method to the distance of an arbitrary path (start position: *q_start_*, goal position: *q_goal_*) is at least equal (If the distances of D and U are the same, when the rewired line segments are on a straight line) to or always shorter than $D_{R}$; as a result, this method is not applied.

Equations (16)–(18) show the path length $D_{A}$ of the path from the start position (root node) of *T_a_* to the last (inserted node) position *q_newA_* and the path length $U_{A}$ when the ‘Triangular-Rewiring’ method has been applied to the path. In addition, it shows that $U_{A}$ is at least equal to or always shorter than $D_{A}$:(16)$$\xi^{\delta_{A}+1}\left(q_{newA}\right)=q_{start},\;k_{\varphi_{A}}=\delta_{A},$$
(17)$$D_{A}=\sum_{j=0}^{\varphi_{A}}\sum_{n={k^{\prime }}_{j}}^{k_{j}}{𝕕}_{n}\left(q_{newA}\right),\;U_{A}=\sum_{j=0}^{\varphi_{A}}{𝕦}_{k_{j}}\left(q_{newA}\right),$$
(18)$$∴D_{A}\geq U_{A}$$

Equations (19)–(21) show the path length $D_{B}$ of the path from the start position (root node) of *T_b_* to the last (inserted node) position *q_newB_* and the path length $U_{B}$ when the ‘Triangular-Rewiring’ method has been applied to the path. In addition, it shows that $U_{B}$ is at least equal to or always shorter than $D_{B}$:(19)$$\xi^{\delta_{B}+1}\left(q_{newB}\right)=q_{goal},\;k_{\varphi_{B}}=\delta_{B},$$
(20)$$D_{B}=\sum_{j=0}^{\varphi_{B}}\sum_{n={k^{\prime }}_{j}}^{k_{j}}{𝕕}_{n}\left(q_{newB}\right),\;U_{B}=\sum_{j=0}^{\varphi_{B}}{𝕦}_{k_{j}}\left(q_{newB}\right),$$
(21)$$∴D_{B}\geq U_{B}$$

Therefore, Equations (16) and (19) can be derived from Equations (3) and (13), Equations (17) and (20) from Equation (14), and Equations (18) and (21) from Equation (15).

As a result, Equations (22) and (23) show that RRT-Connect with the proposed ‘Triangular-Rewiring’ method is at least the same or better in terms of path length than the RRT-Connect algorithm without the method
(22)$$D_{R}=D_{A}+D_{B}+D\left(q_{newA},\;q_{newB}\right),\;U_{R}\leq U_{A}+U_{B}+D\left(q_{newA},\;q_{newB}\right),$$
(23)$$∴D_{R}\geq D_{A}+D_{B}\geq U_{A}+U_{B}\geq U_{R}$$

$D_{R}$ (Equation (4)), which refers to the path length of the RRT-Connect algorithm path without the ‘Triangular-Rewiring’ method, is represented by the sum of the distance $D_{A}$ of the partial path *P_a_* (Equation (17)), the distance $D_{B}$ of the partial path *P_b_* (Equation (20)), and the distance $D\left(q_{newA},\;q_{newB}\right)$ between *q_newA_* and *q_newB_* as shown in Equation (22).

$U_{R}$ (Equation (14)), which refers to the path length of the RRT-Connect algorithm path with the ‘Triangular-Rewiring’ method, is equal to or shorter than the sum of the distance $U_{A}$ of the partial path *P_a_* for the RRT-Connect (Equation (17)), the distance $U_{B}$ of the partial path *P_b_* (Equation (20)), and the distance $D\left(q_{newA},\;q_{newB}\right)$ between *q_newA_* and *q_newB_* as shown in Equation (22).

Here, Equation (23) shows that $U_{R}$ is at least equal to or shorter than $D_{R}$ in the RRT algorithm summarized in Equation (15), and it is used efficiently in the RRT-Connect algorithm.

### 4.3. Pseudocode of Proposed Extend Method for the Improved RRT-Connect Algorithm

This section introduces the ‘Extend’ method in the proposed triangular inequality-based RRT-Connect algorithm. This proposed ‘Extend’ method (A5) replaces the ‘Extend’ method (A3) in the pseudocode of the RRT-Connect algorithm (A2).

Algorithm 5 is the application of the ‘Triangular-Rewiring’ method (A4) to the original ‘Extend’ method (A2) of the RRT-Connect algorithm. Compared to the original ‘Extend’ method, the part where a node is newly inserted in the tree in lines 9 and 16 is inserted through the ‘Triangular-Rewiring’ method. Other than that, the contents are the same as the original ‘Extend’ method.
**Algorithm 5** Pseudocode of the Proposed ‘Extend’ Method for the RRT-Connect Algorithm**Input:***T_a_* ← Tree *T_a_* from *RRT-Connect**T_b_* ← Tree *T_b_* from *RRT-Connect**q_newB_* ← Position *q_newB_* from *RRT-Connect**q_rand_* ← Position *q_rand_* from *RRT-Connect**λ* ← Step Length *λ* from **RRT-Connect***C* ← Position Set *C* from *RRT-Connect***Output:***f_trap_* ← Result of Boolean *f_trap_**T_a_* ← Result of Tree *T_a_* **//Return by Reference***T_b_* ← Result of Tree *T_b_* **//Return by Reference***q_newB_* ← Result of Position *q_newB_* **//Return by Reference****Initialize:***f_trap_* ← ***False*****Begin***Extend***Procedure from***RRT-Connect*1*q_near_* ← **Find** Position of Nearest Node in *T_a_* from *q_rand_*2**If Not***isInside*(*q_near_, q_rand_, λ*) **then**3 *q_newA_* ← Position of Intersection Point between Line Segment connecting *q_rand_* and *q_near_*, and Circle with Radius *λ* centered at *q_near_* **//2D:** Circle**, 3D:** Sphere**, …**4**Else**5 *q_newA_* ← *q_rand_*6**If***isTrapped*(*q_newA_, q_near_, C*) **then**7 *f_trap_* ← ***True***8**Else**9 *T_a_* ← *triangularRewiring*(*q_newA_, q_near_, T_a_, C*)10 *q_near_* ← **Find** Position of Nearest Node in *T_b_* from *q_newA_*11 **If**
*isInside*(*q_near_, q_newA_, λ*) **then**12  *q_newB_* ← *q_near_*13 **Else**14  *q_newB_* ← Position of Intersection Point between Line Segment connecting *q_newA_* and *q_near_*, and Circle with Radius *λ* centered at *q_near_* **//2D:** Circle**, 3D:** Sphere**, …**15  **While Not**
*isTrapped*(*q_newB_, q_near_, C*) **do**16   *T_b_* ← *triangularRewiring*(*q_newB_, q_near_, T_b_, C*)17   **If Not**
*isInside*(*q_newA_, q_newB_, λ*) **then**18    *q_near_* ← *q_newB_*19    *q_newB_* ← Position of Intersection Point between Line Segment connecting *q_newA_* and *q_near_*, and Circle with Radius *λ* centered at *q_near_* **//2D:** Circle**, 3D:** Sphere**, …**20   **Else**21    ***Break*****End***Extend***Procedure from***RRT-Connect*

Figure 7 shows the application of the ‘Triangular-Rewiring’ method to Figure 3, which shows the ‘Extend’ method of the RRT-Connect algorithm. In *T_a_*, *q_newA_* and *q_start_* are rewired and *q_near_* and *q_goal_*, and *q_newB_* and *q_goal_* are rewired sequentially in the process of extending from *T_b_* to *T_a_*.

### 4.4. Pseudocode of the Proposed Connect Method for the RRT-Connect Algorithm

This section introduces the ‘Connect’ method in the proposed triangular inequality-based RRT-Connect algorithm. This proposed ‘Connect’ method (A6) replaces the ‘Connect’ method (A4) in the pseudocode of the RRT-Connect algorithm (A2).

Algorithm 6 is an application of the ‘Triangular-Rewiring’ method (A4) to the ‘Connect’ method (A3) of the RRT-Connect algorithm. Compared to the original ‘Connect’ method, it has been changed to apply the method to the merged tree by considering the ‘Triangular-Rewiring’ method when merging the path, which is in lines 5–10. Other than that, the contents are the same as the original ‘Connect’ method.
**Algorithm 6** Pseudocode of the Proposed ‘Connect’ Method for the RRT-Connect Algorithm**Input:***P_reach_* ← Path *P_reach_* from *RRT-Connect**T_a_* ← Tree *T_a_* from *RRT-Connect**T_b_* ← Tree *T_b_* from *RRT-Connect**q_newB_* ← Position *q_newB_* from *RRT-Connect**λ* ← Step Length *λ* from *RRT-Connect***Output:***f_reach_* ← Result of Boolean *f_reach_**P_reach_* ← Result of Path *P_merged_* **//Return by Reference****Initialize:***f_reach_* ← ***False*****Begin***Connect***Procedure from***RRT-Connect*1**If***isInside*(*q_newA_, q_newB_, λ*) **then**2 *P_a_* ← Path from Root Node [*q_start_*] to Last Inserted Node [*q_newA_*] in *T_a_*3 *P_b_* ← Path from *q_newB_* to Root Node [*q_goal_*] in *T_b_*4 *P_connect_* ← Path from Last Inserted Node [*q_newA_*] in *T_a_* to *q_newB_* in *T_b_*5 *T_merged_* ← Tree Structure with **Merge Path**
*P**_a_* to *P_b_* via *P_connect_***//1st Insert:**
*q_start_*, …, ***n*****-th Insert:**
*q_newA_*, **(*****n***
**+ 1)-th Insert:**
*q_newB_*, …, **Last Insert:**
*q_goal_*
**to**
*T_merged_*6 **For**
*i* ← Inserted Index of *q_newA_* in *T_merged_*
**to** (Number of Node in *T_merged_*) – 1 **do**7  *q_new_* ← (*i* – 1)-th Inserted Node in *T_merged_*8  *q_near_* ← *i*-th Inserted Node in *T_merged_*9  *T_merged_* ← *triangularRewiring*(*q_new_, q_near_, T_merged_, C*)10 *P_merged_* ← Path from Root Node [*q_start_*] to Last Inserted Node [*q_goal_*] in *T_merged_*11 *f_reach_* ← ***True*****End***Connect***Procedure from***RRT-Connect*

When paths *P_a_* and *P_b_* merge in a tree structure of line 5, nodes on the path are inserted in the order of *P_a_*, *P_connect_*, and *P_b_* in the merged tree *T_merged_*. That is, in *T_merged_*, the root node becomes *q_start_*, and when the *n*-th inserted node at a certain point is *q_newA_*, which is the last inserted node of *T_a_*, the (*n* + 1)-th inserted node becomes *q_newB_*, which is the last inserted node of *T_b_*. In addition, the last inserted node of *T_merged_* becomes *q_goal_*.

Then, the ‘Triangular-Rewiring’ method is applied to this *T_merged_*. Since it is applied to the tree itself, it determines whether rewiring is possible for all nodes inserted in the tree, and rewires and updates the tree if possible. However, since each node from *T_a_* to *T_b_* is inserted into *T_merged_*, it is not necessary to rewire *T_a_* for which the ‘Triangular-Rewiring’ process has already been performed. Therefore, the ‘Triangular-Rewiring’ process proceeds in the direction of *T_b_* from the *q_newA_* sequence inserted in *T_merged_*. Here, if *q_newA_* is the *i*-th inserted node, the first node pair to be determined is the (*i* − 1)-th node *q_new_* (as *q_child_*) and *i*-th node *q_near_* (as *q_parent_*). When all nodes inserted in *T_merged_* have been determined, the tree structure *T_merged_* is converted into the path *P_merged_* and the method terminates (True).

Figure 8 shows the ‘Triangular-Rewiring’ method applied to Figure 4, which shows the ‘Connect’ method of the RRT-Connect algorithm. When the paths *P_a_* and *P_b_* created from the trees *T_a_* and *T_b_* are merged and the ‘Triangular-Rewiring’ method has been applied (assuming there is no obstacle between *q_start_* and *q_goal_*), the result is *P_merged_* in which *q_start_* and *q_goal_* are connected with a straight line.

### 4.5. Process of the Proposed Triangular Inequality-Based RRT-Connect Algorithm

Figure 9 in this section shows the path-planning process of the proposed algorithm by applying the ‘Triangular-Rewiring’ method to the ‘Extend’ and ‘Connect’ methods of the RRT-Connect algorithm.

Figure 9 shows planning a path from the start position *q_start_* to the goal position *q_goal_* through the proposed algorithm, as shown in Figure 9a.

In Figure 9b, the first random sample is generated at position *q_rand_* and *q_newA_* is created at a position separated by the length of *λ* from *q_start_* in the direction of the position, and *q_newA_* is extended once by the length of *λ* in the direction of *q_newA_* from *q_goal_*. At this time, since there is no intermediate node between *q_newA_* and *q_start_*, the ‘Triangular-Rewiring’ process is skipped.

In Figure 9c, a second random sample is generated at the *q_rand_* position, and in the direction of the position, *q_newA_* is updated at a location separated by *λ* length from the nearest node *q_near_* in the tree and rewired between *q_newA_* and *q_goal_*. In this case, since the tree on the opposite side collides with an obstacle to extend in the *q_newA_* direction, the ‘Extend’ process is skipped. In addition, it is assumed that *Swap* occurs between *T_a_* with initial *q_start_* as the root node and *T_b_* with initial *q_goal_* as the root node between each figure.

In Figure 9d, as shown in Figure 9c, a third random sample is created at the *q_rand_* position and at a position separated by the length of *λ* in the position direction, at the node *q_near_* that is nearest among nodes in the tree in the position direction, It shows updating *q_newA_* to a position that is the length of *λ* and rewires it between *q_newA_* and *q_start_*. Here, since it also collides with an obstacle to extend in the direction of *q_newA_* from the tree on the opposite side, the ‘Extend’ process is skipped.

In Figure 9e, the fifth random sample is generated at the *q_rand_* position and *q_newA_* is located at a position separated by the length of *λ* in the direction of the position, and *q_newA_* is also at a position separated by the length of *λ* from the nearest node *q_near_* among nodes in the tree toward the position. It is shown when updating that *q_newA_* merges into one tree through the ‘Connect’ process because *q_newA_* is within range of the center of *q_newB_* and the radius of *λ*. It is assumed that the fourth random sample between Figure 9d,e is generated inside the obstacle, so the *q_newA_* generation process is skipped. Figure 9f shows the result of path *R* created as a merged tree by ‘Connect’ as shown in Figure 9e.

## 5. Experimental Results

To verify the performance of the proposed triangular inequality-based RRT-Connect algorithm in this paper, the RRT algorithm, the RRT-Connect algorithm, and the proposed algorithm are compared in various environment maps shown in the experimental environment through the simulator.

Each algorithm was implemented based on the pseudocode (A1–9) shown Section 3 and Section 4 (For the RRT algorithm, refer to the pseudocode (AA1) in Appendix A), and the performance measures used for comparison of various algorithms are Number of samples (samples), Path length (pixels), and Planning time (milliseconds). Each performance measure is experimented with 50 trials from the same start point to the same goal point until the first path has been found). Among the performance measures, as the number of samples decreases, the cost of recalculation in a dynamic environment also decreases, and the path length is a measure of the optimality of the path-planning algorithm. In addition, the Step length (*λ*) is 30 pixels.

### 5.1. Experimental Environment

This section introduces the environment map used in the simulation and the simulator used in the simulation with the computer’s performance.

Figure 10 shows the eight environmental maps used in this experiment. The green circle (S) indicates the start point, the purple circle (G) indicates the goal point, and the black polygon on the yellow (blue in the analysis of the experimental results) border indicates to the obstacle. All maps are 600 (horizontal) * 600 (vertical) pixels.

Many environmental maps were considered and used to verify the performance of various path-planning algorithms including the RRT algorithm, [23,24,25,26]. Which environment map to use is important because the expected performance measure varies depending on the obstacles’ placement and shape among other properties.

In this paper, to check the proposed algorithm’s performance, the eight maps shown in Figure 10 were benchmarked in the experimental environment of the paper [27] proposed by Jihee Han in 2017, and each map is expected to have the following features:

Map 1 in Figure 10a seems to be an environment in which it is easy to verify the completeness of the path-planning algorithm. Map 2 in Figure 10b seems to be an environment in which it is also easy to verify the completeness of the path-planning algorithm, and the environment is mainly used to show the solution for the Local Minima problem [28] in the artificial potential field algorithm [26]. Map 3 in Figure 10c seems to be an environment in which it is easy to verify the optimality and completeness of the path-planning algorithm and is an environment that is unfavorable to random sampling path-planning algorithms such as the RRT algorithm. Map 4 in Figure 10d seems to be an environment in which it is easy to verify the optimality and the planning time for the path-planning algorithm, and the cell decomposition algorithm, which increases the computation cost as the angle of obstacle increases, is an unfavorable environment [29]. Map 5 in Figure 10e seems to be an environment in which it is also easy to verify the optimality and planning time of the path-planning algorithm; for the same reason as Map 4, the cell decomposition algorithm is an unfavorable environment. Map 6 in Figure 10f seems to be an environment in which it is easy to verify the optimality, completeness, and planning time of the path-planning algorithm, and it is an environment for comprehensively evaluating the performance. Map 7 in Figure 10g seems to be an environment in which it is easy to verify the completeness and optimality of the path-planning algorithm, and for the same reason as Map 2, it is the environment used in the Artificial Potential Field algorithm. Lastly, Map 8 in Figure 10h seems to be an environment in which it is easy to verify the completeness and planning time of the path-planning algorithm and yet is unfavorable to random sampling path-planning algorithms such as the RRT algorithm.

Since random sampling path-planning algorithms such as the RRT algorithm rely on probabilistic completeness, the number of samples and the planning time are significantly increased as long as there are narrow or fewer entrances for directions to the goal.

Table 1 shows the specifications of the computer used in the simulation. The simulator was developed in C# language (Microsoft Visual Studio Community 2019 version 16.1.6; Microsoft. NET Framework version 4.8.03752), and except for the visual part, only a single thread was used for the calculation. Differences in planning time may occur depending on the computer’s performance capability.

### 5.2. Experimental Results and Analysis for Each Map

This section checks the experimental results (on average, the number of samples, path length, and planning time) of each algorithm: RRT, RRT-Connect, the proposed algorithm in the eight environment maps (Figure 10) presented in the experimental environment. Each map shows a figure of the path-planning result (of one trial) for each algorithm (Figures 11–18) and the experimental results for the performance measure are shown numerically in a table (Tables 2–9). The figure for each algorithm is for one trial rather than the average of repeated trials and it may differ from the performance measure both visually and by the average numerical performance measure of the repeated trials shown in the table. In particular, the number of samples differs greatly.

The values shown in Tables 2–9 can be expressed as Equations (24) and (25) as
(24)$$A_{cmp}\left(i\right)\;=\overset{T}{\underset{k=0}{\sum }}a_{cmp{}_{k}}\left(i\right)/T$$

Here, $A_{cmp}\left(i\right)$ refers to the performance value of each algorithm shown in Tables 2–9, *cmp* is the algorithm to be compared, *i* is the index of the environment map (*X*-axis in Figures 19, 20 and 21b, *k* is the repeat index, and *T* is the number of repeats ($a_{cmp}{}_{k}\left(i\right)$ is the value of the performance measure *a* for the *k*-th implementation of the *cmp* algorithm in Map *i*). Fifty repetitions are performed for the experiment in this paper. That is, Equation A shows the average value of the performance when it is repeated *T* times to check the performance of a certain algorithm in Map *i*,
(25)$$∴x_{cmp}\left(i\right)\;=A_{cmp}\left(i\right)/A_{RRT}\left(i\right)$$

Here, $x_{cmp}\left(i\right)$ refers to the *Y*-axis in Figures 19, 20 and 21a and *A* is the value of the corresponding performance measure of the algorithm to be compared ($A_{RRT}$ is the value of the RRT algorithm).

In each path-planning result figure, the white circles indicate nodes on the path and the yellow line segments indicate edges between nodes. The gray circles and segments are paths (trees) that have been excluded during path planning. In each path-planning result table, based on 100% of the RRT algorithm for each performance measure, the difference is indicated along with the value of the corresponding performance measure unit.

Figure 11 shows the path-planning results of Map 1 among the environmental maps for each algorithm. Visually, the number of samples looks similar to the RRT-Connect algorithm in Figure 11b and the proposed algorithm in Figure 11c is comparable to the RRT algorithm in Figure 11a, and the path length looks similar for all three algorithms.

Table 2 shows the path-planning results (after repeating the trial 50 times) in Map 1 for each algorithm. The average number of samples is the smallest in RRT-Connect algorithm at 60%, and the proposed algorithm is 68% compared to the RRT algorithm, which is 8% less efficient than the RRT algorithm compared to the RRT-Connect algorithm. The average path length is shortest for the proposed algorithm at 89% compared to the RRT algorithm, with little difference in the RRT-Connect algorithm at 100%, and 11% less efficient than the proposed algorithm. The average planning time is the shortest for the RRT-Connect algorithm at 58% compared to the RRT algorithm, and the proposed algorithm is 83% compared to the RRT algorithm, i.e., 15% less efficient than the RRT algorithm.

Figure 12 shows the path-planning results of Map 2 among the environmental maps for each algorithm. Visually, the number of samples looks similar for the RRT-Connect algorithm in Figure 12b and the proposed algorithm in Figure 12c compared to the RRT algorithm in Figure 12a, and the path length looks shortest for the proposed algorithm.

Table 3 shows the path-planning result (after repeating the trials 50 times) in Map 2 for each algorithm. The average number of samples is smallest in the RRT-Connect algorithm at 37%, and the proposed algorithm is 42% compared to the RRT algorithm, which is 5% less efficient than RRT algorithm compared to the RRT-Connect algorithm. The average path length of the proposed algorithm is the shortest at 81% compared to the RRT algorithm, while the RRT-Connect algorithm is 98%, which is 17% less efficient than the RRT algorithm compared to the proposed algorithm. The average planning time for the proposed algorithm and the RRT-Connect shows the same performance as the RRT algorithm.

Figure 13 shows the path planning results of Map 3 among the environmental maps for each algorithm. Visually, the number of samples looks similar for the RRT-Connect algorithm in Figure 13b and the proposed algorithm in Figure 13c compared to the RRT algorithm in Figure 13a, and the path length looks shortest for the proposed algorithm.

Table 4 shows the result (after repeating the trial 50 times) of path planning in Map 3 for each algorithm. The average number of samples is smallest in the RRT-Connect algorithm at 75%, and the proposed algorithm is 77% compared to the RRT algorithm, which is 2% less efficient than the RRT algorithm compared to the RRT-Connect algorithm. The average path length of the proposed algorithm is the shortest at 77% compared to the RRT algorithm and the RRT-Connect algorithm is 97%, which is 20% less efficient than the RRT algorithm compared to the proposed algorithm. The average planning time is smallest for the RRT-Connect algorithm at 35%, and the proposed algorithm is 36% compared to the RRT algorithm, which is 1% less efficient than the RRT algorithm compared to the RRT-Connect algorithm.

Figure 14 shows the path planning results of Map 4 among the environmental maps for each algorithm. Visually, the number of samples looks smallest for the RRT-Connect algorithm in Figure 14b compared to the others and the path length looks shortest for the proposed algorithm in Figure 14c.

Table 5 shows the result (after repeating the trial 50 times) of path planning in Map 4 for each algorithm. The average number of samples is smallest in the RRT-Connect algorithm at 10%, and the proposed algorithm is 11% compared to the RRT algorithm, which is 1% less efficient than the RRT algorithm compared to the RRT-Connect algorithm. The average path length of the proposed algorithm is the shortest at 75% compared to the RRT algorithm and the RRT-Connect algorithm is 83%, which is 8% less efficient than the RRT algorithm compared to the proposed algorithm. The average planning time is not different by 100% compared to the RRT algorithm, and the proposed algorithm is 133% compared to the RRT algorithm, i.e., 33% less efficient than the others.

Figure 15 shows the path planning results of Map 5 among the environmental maps for each algorithm. Visually, the number of samples looks similar for the RRT-Connect algorithm in Figure 15b and the proposed algorithm in Figure 15c compared to the RRT algorithm in Figure 15a, and the path length looks similar for the RRT-Connect algorithm and the proposed algorithm.

Table 6 shows the results (after repeating the trial 50 times) of path planning in Map 5 for each algorithm. The average number of samples is smallest in RRT-Connect algorithm at 18%, and the proposed algorithm is 20% compared to the RRT algorithm, which is 9% less efficient than the RRT algorithm compared to the RRT-Connect algorithm. The average path length of the proposed algorithm is the shortest at 84% compared to the RRT algorithm and the RRT-Connect algorithm is 106%, which is 22% less efficient compared to the proposed algorithm. The average planning time for the proposed algorithm and the RRT-Connect algorithm is 15% over the RRT algorithm, showing the same performance.

Figure 16 shows the path-planning results of Map 6 among the environmental maps for each algorithm. Visually, the number of samples looks smallest for the proposed algorithm in Figure 16c compared to others, and the path length looks shortest for the proposed algorithm.

Table 7 shows the result (after repeating the trial 50 times) of path planning in Map 6 for each algorithm. The average number of samples is smallest in the proposed algorithm at 26% and the RRT-Connect algorithm is 34% compared to the RRT algorithm, which is 8% less efficient than RRT algorithm compared to the proposed algorithm. The average path length of the proposed algorithm is the shortest at 75% compared to the RRT algorithm, and the RRT-Connect algorithm is 88%, which is 13% less efficient than the proposed algorithm. The average planning time is smallest in the proposed algorithm at 44%, and the RRT-Connect is 67% compared to the RRT algorithm, which is 23% less efficient than the RRT algorithm compared to the proposed algorithm.

Figure 17 shows the path planning results of Map 7 among the environmental maps for each algorithm. Visually, the number of samples looks smallest for the proposed algorithm in Figure 17c compared to others, and the path length looks shortest for the proposed algorithm.

Table 8 shows the result (after repeating the trial 50 times) of path planning in Map 7 for each algorithm. The average number of samples is smallest in RRT-Connect algorithm at 54%, and the proposed algorithm is 56% compared to the RRT algorithm, which is 2% less efficient than RRT algorithm compared to the RRT-Connect algorithm. The average path length of the proposed algorithm is shortest at 75% compared to the RRT algorithm and the RRT-Connect algorithm is 96%, which is 21% less efficient compared to the proposed algorithm. The average planning time is smallest in the proposed algorithm at 60%, and RRT-Connect is 80% compared to the RRT algorithm, which makes it 20% less efficient than the RRT algorithm compared to the proposed algorithm.

Figure 18 shows the path planning results of Map 8 among the environmental maps for each algorithm. Visually, the number of samples looks similar for the RRT-Connect algorithm in Figure 18b and the proposed algorithm in Figure 18c compared to the RRT algorithm in Figure 18a, and the path length looks shortest for the proposed algorithm.

Table 9 shows the result (after repeating the trial 50 times) of path planning in Map 8 for each algorithm. The average number of samples is smallest in the proposed algorithm at 17%, and the RRT-Connect algorithm is 18% compared to the RRT algorithm, which is 1% less efficient than RRT algorithm compared to the proposed algorithm. The average path length of the proposed algorithm is the shortest at 84% compared to the RRT algorithm, and the RRT-Connect algorithm is 98%, which is 14% less efficient compared to the proposed algorithm. The average planning time of the proposed algorithm and the RRT-Connect algorithm is 3% over the RRT algorithm, showing the same performance.

### 5.3. Experimental Results and Analysis in Total

This section comprehensively presents the experimental results (on average, number of samples, path length, and planning time) for each algorithm: RRT, RRT-Connect, and the proposed triangular inequality-based RRT-Connect algorithm, in the eight environmental maps (Figure 10) shown in Section 5.2.

Figures 19a–21a show the performances of the RRT-Connect algorithm and the proposed algorithm when the RRT algorithm’s performance is set to 100% for each environment map. The (b) of each figure shows the performance average of all environment maps for each algorithm. The values shown in (a) of Figures 19–21 can be expressed as in Equations (24) and (25) and the values shown in (b) can be expressed as Equation (26)
(26)$$X_{cmp}=\overset{M}{\underset{i=0}{\sum }}x{\left(i\right)}_{cmp}/M$$

Here, $X_{cmp}$ refers to the *Y*-axis in (b) of Figures 19–21 and *M* is the number of environment maps used in the experiment. The experiment in this paper includes eight maps. That is, Equation (26) shows the average value of *I* for all maps in Equation (25).

Figure 19 shows the average number of samples [%] compared with the RRT algorithm for Maps 1–8 (after repeating the trial 50 times) and the average number of samples [%] compared with the average result of each algorithm for each map (after repeating the trials 50 times) when the result of RRT algorithm is considered 100%.

As shown in Figure 19b, the average number of samples for all environment maps was 38% less in the RRT-Connect algorithm and 40% less in the proposed algorithm compared to the RRT algorithm. The proposed algorithm is 2% less efficient than the RRT-Connect algorithm.

Table 10 is the data table of Figure 19a. The proposed algorithm shows better performance than the RRT-Connect algorithm for Maps 6 and 8 and the RRT-Connect algorithm shows better performance than the proposed algorithm in Maps 1–5 and 7. However, the difference is not significant for most of the maps, such as showing a 2% difference from the map average. There are cases in which the proposed algorithm is 1–8% better than the RRT-Connect algorithm and there are cases in which the RRT-Connect algorithm is 1–8% better than the proposed algorithm.

Figure 20 shows the average path length [%] compared to the RRT algorithm for Maps 1–8 (after repeating the trials 50 times), and the average path length (%) compared with the average result of each algorithm for each map (again after repeating the trials 50 times) where the result of the RRT algorithm was considered as 100%.

As shown in Figure 20b, the average path length for all environment maps was 96% less in the RRT-Connect algorithm and 80% less in the proposed algorithm compared to the RRT algorithm. The proposed algorithm is 16% more efficient than the RRT-Connect algorithm.

Table 11 is the data table of Figure 20a. The proposed algorithm shows better performance than the RRT-Connect algorithm for all maps. The proposed algorithm is 8–21% better than the RRT-Connect algorithm.

Figure 21 shows the average planning time [%] compared to the RRT algorithm for each map (after repeating the trials 50 times), and the average planning time (%) compared with the average result of each algorithm for each map (after repeating the trials 50 times) when the result of RRT algorithm is considered 100%.

As shown in Figure 21b, the average planning time for all environment maps was 51% less in the RRT-Connect algorithm and 53% less in the proposed algorithm compared to the RRT algorithm. The proposed algorithm was 2% less efficient than the RRT-Connect algorithm.

Table 12 is the data table of Figure 21a. The proposed algorithm shows the same or better performance for Maps 2 and 5–8 than the RRT-Connect algorithm. It shows worse performance for Maps 1, 3, and 4 than the RRT-Connect algorithm. However, most of the maps show no significant difference, such as showing a 2% difference from the map average. There are cases in which the proposed algorithm is 20–23% better than the RRT-Connect algorithm and there are cases where the RRT-Connect algorithm is 1–33% better than the proposed algorithm.

## 6. Conclusions

In this paper, we proposed a triangular inequality-based RRT-Connect algorithm using triangular inequality principles to overcome the limitations in the optimality of the RRT-Connect algorithm. We verified the validity of the ‘Triangular-Rewiring’ method based on the triangular inequality principle and applied it to the RRT-Connect algorithm to bring it closer to the optimum. In addition, to check performance indicators such as the number of samples for finding the first path, path length, and planning time of the proposed algorithm, we compared between the RRT and RRT-Connect algorithms across a total of eight environments through simulation. On average, the proposed algorithm showed 20% better efficiency than the RRT algorithm and 16% better efficiency than the RRT-Connect algorithm in path length and 47% better efficiency than the RRT algorithm in planning time but 2% worse efficiency than the RRT-Connect algorithm. In conclusion, the proposed algorithm showed shorter paths than the RRT-Connect algorithm with a similar number of samples and planning time.

However, one of the limitations of the proposed algorithm is the Kinodynamic planning problem [17]. When the intermediate node disappears by ‘Triangular-Rewiring’ method, a non-differentiable piecewise linear section with sharp corner may occurs, which cause a problem related with the kinematic constraint of the robot.

## Figures and Tables

**Figure 1 sensors-21-00333-f001:**
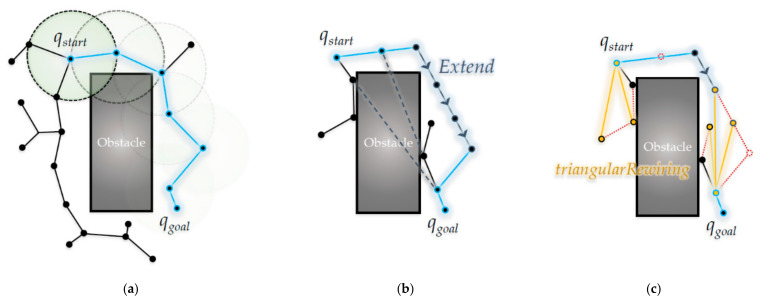
Overview of the algorithms in this paper: (**a**) RRT; (**b**) RRT-Connect; (**c**) the proposed algorithm.

**Figure 2 sensors-21-00333-f002:**
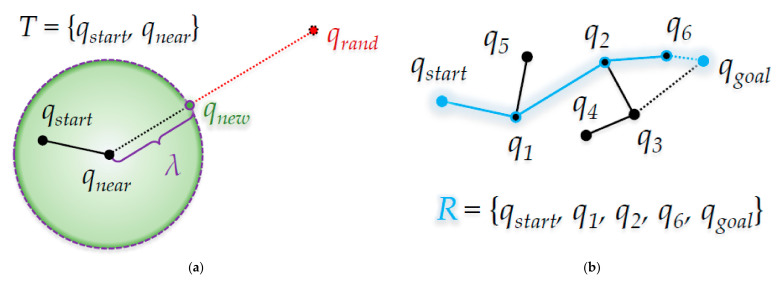
RRT algorithm: (**a**) Process when *q_new_* is created; (**b**) After the random sampling has ended.

**Figure 3 sensors-21-00333-f003:**
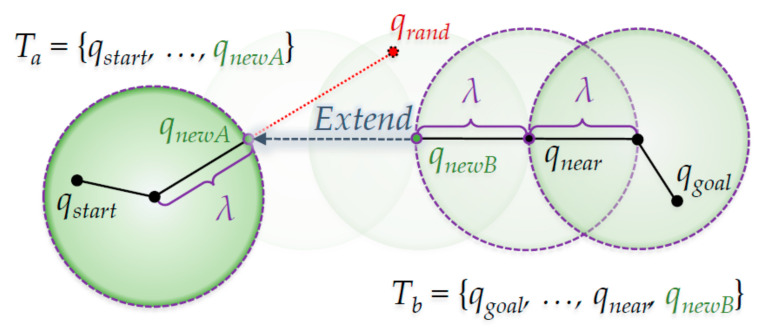
The ‘Extend’ method from RRT-Connect algorithm.

**Figure 4 sensors-21-00333-f004:**
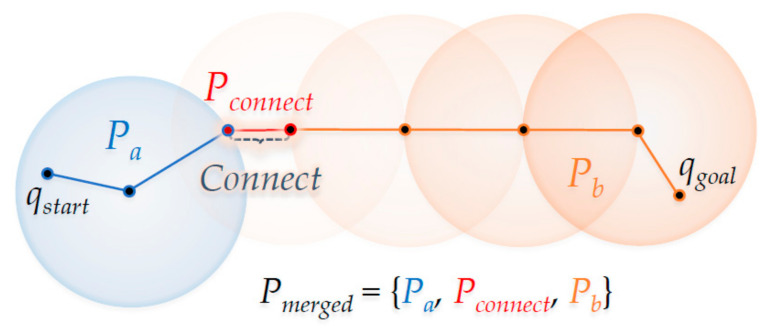
The ‘Connect’ method from the RRT-Connect algorithm.

**Figure 5 sensors-21-00333-f005:**
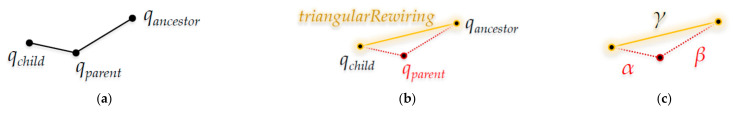
Abstract process of the ‘Triangular-Rewiring’ method: (**a**) Example tree; (**b**) After rewiring between *q_child_* and *q_ancestor_*; (**c**) At this time, *α* is the distance between *q_child_* and *q_parent_*, *β* is the distance between *q_parent_* and *q_ancestor_*, and *γ* is the distance between *q_child_* and *q_ancestor_*.

**Figure 6 sensors-21-00333-f006:**
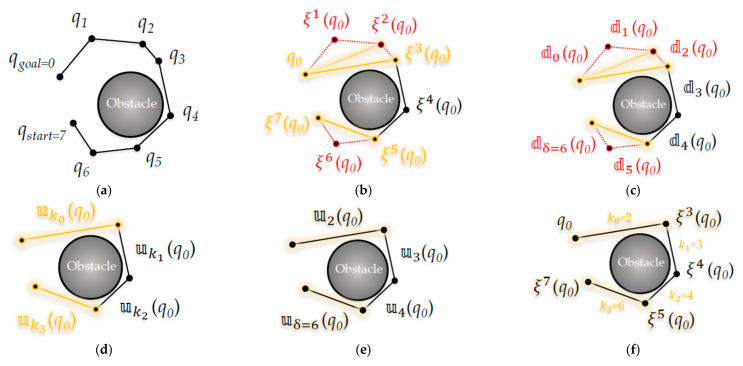
Detailed process of the ‘Triangular-Rewiring’ method: (**a**) Each node *q* for index *i* (at this time, *q_start_* is same as *q*_7_ and *q_goal_* is same as *q*_0_); (**b**) Represent each node using the *n*-th ancestor $\xi^{n}$
of *q*_0_; (**c**) Each distance ${𝕕}_{n}$ between the *n*-th and (*n* + 1)-th ancestor nodes of *q*_0_; (**d**) When the ‘Triangular-Rewiring’ method is applied and rewired by distance ${𝕦}_{k_{j}}$; (**e**) Represent as the value of *k_j_*; (**f**) Represent each node by the *n*-th ancestor $\xi^{n}$ of *q*_0_ after method is applied.

**Figure 7 sensors-21-00333-f007:**
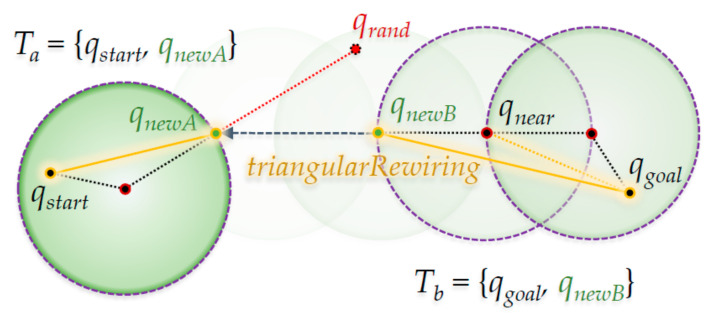
Proposed ‘Extend’ method for the RRT-Connect algorithm.

**Figure 8 sensors-21-00333-f008:**
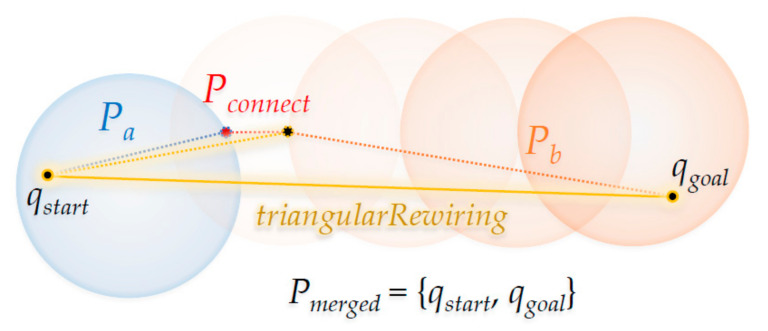
Proposed ‘Connect’ method for the RRT-Connect algorithm.

**Figure 9 sensors-21-00333-f009:**
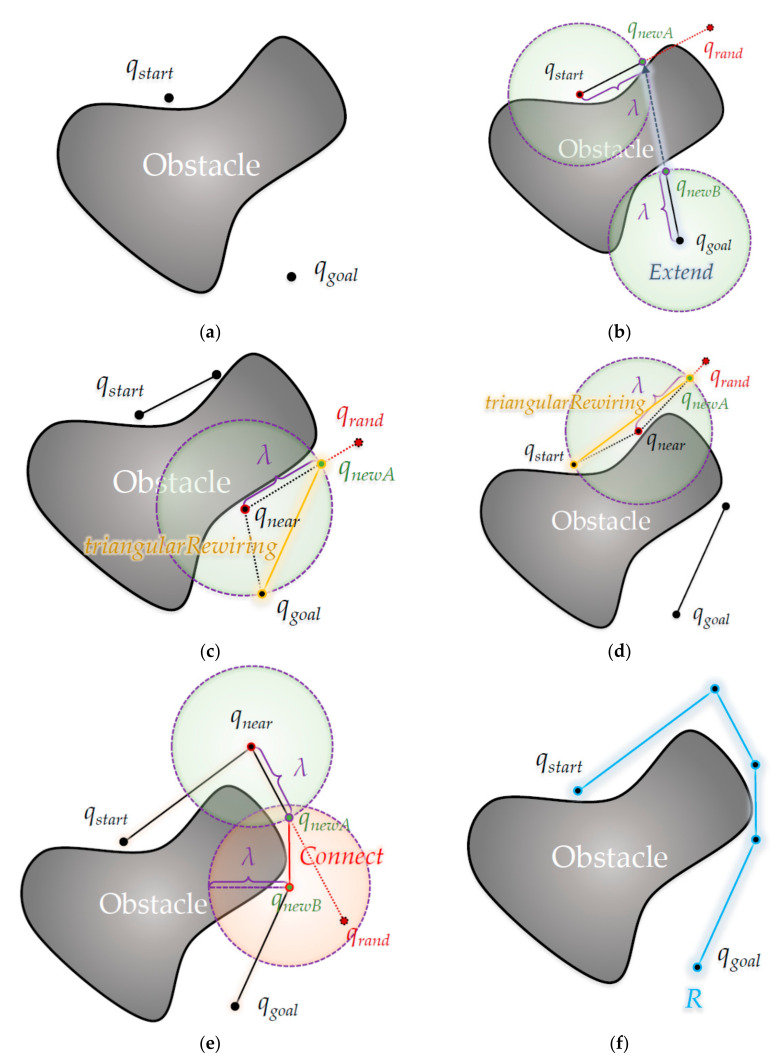
Detailed process of the proposed algorithm: (**a**) Start position *q_start_* from tree *T_a_* and goal position *q_goal_* from tree *T_b_*; (**b**) Create *q_newA_* nearest to *T_a_* from 1^st^ random sampling position *q_rand_* and create *q_newB_* from *q_goal_* nearest to *T_b_*; (**c**) Create new *q_newA_* from *q_near_* nearest to *T_b_* from the second random sampling position *q_rand_* and rewire between *q_newA_* and *q_goal_* the ancestor of the *q_newA_*; (**d**) Create a new *q_newA_* from *q_near_* nearest to *T_a_* from the third random sampling position *q_rand_* and rewire between *q_newA_* and *q_start_* with the ancestor of *q_newA_*; (**e**) Create new *q_newA_* from *q_near_* nearest to *T_a_* from the fifth random sampling position *q_rand_* and connect between *q_newA_* and *q_newB_* nearest to *T_b_* from *q_newA_*; (**f**) Result of Path *R* from *q_start_* to *q_goal_*.

**Figure 10 sensors-21-00333-f010:**
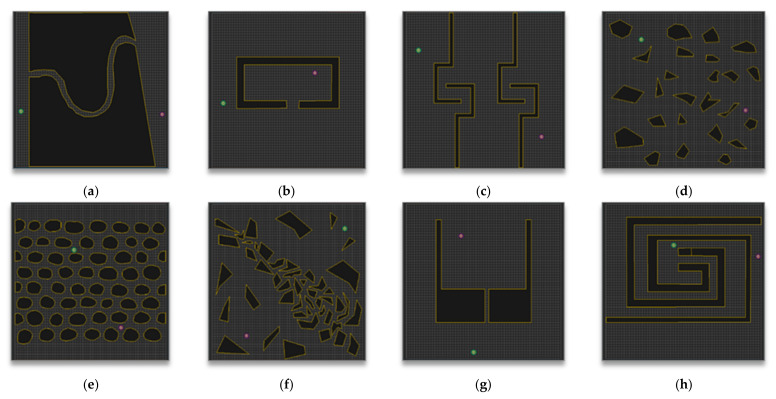
Maps for the experiment: (**a**) Map 1; (**b**) Map 2; (**c**) Map 3; (**d**) Map 4; (**e**) Map 5; (**f**) Map 6; (**g**) Map 7; (**h**) Map 8.

**Figure 11 sensors-21-00333-f011:**

Experimental result of Map 1: (**a**) RRT; (**b**) RRT-Connect; (**c**) the proposed algorithm.

**Figure 12 sensors-21-00333-f012:**
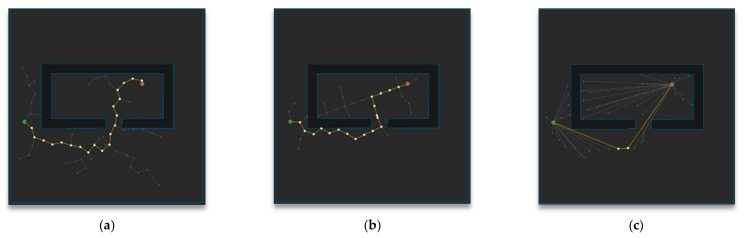
Experimental results of Map 2: (**a**) RRT; (**b**) RRT-Connect; (**c**) the proposed algorithm.

**Figure 13 sensors-21-00333-f013:**

Experimental result of Map 3: (**a**) RRT; (**b**) RRT-Connect; (**c**) the proposed algorithm.

**Figure 14 sensors-21-00333-f014:**

Experimental result of Map 4: (**a**) RRT; (**b**) RRT-Connect; (**c**) the proposed algorithm.

**Figure 15 sensors-21-00333-f015:**

Experimental result of Map 5: (**a**) RRT; (**b**) RRT-Connect; (**c**) the proposed algorithm.

**Figure 16 sensors-21-00333-f016:**

Experimental result of Map 6: (**a**) RRT; (**b**) RRT-Connect; (**c**) the proposed algorithm.

**Figure 17 sensors-21-00333-f017:**
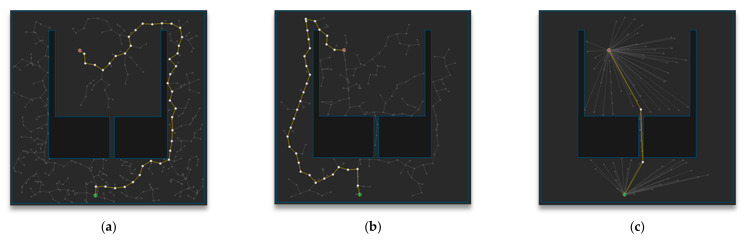
Experimental result of Map 7: (**a**) RRT; (**b**) RRT-Connect; (**c**) the proposed algorithm.

**Figure 18 sensors-21-00333-f018:**

Experimental result of Map 8: (**a**) RRT; (**b**) RRT-Connect; (**c**) the proposed algorithm.

**Figure 19 sensors-21-00333-f019:**
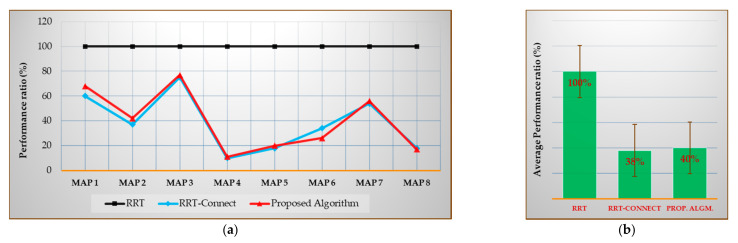
Experimental results in total for the average number of samples (for first path finding): (**a**) result of each map compared with the RRT algorithm ($x_{cmp}\left(i\right)$); (**b**) average result compared with the RRT algorithm ($X_{cmp}$).

**Figure 20 sensors-21-00333-f020:**
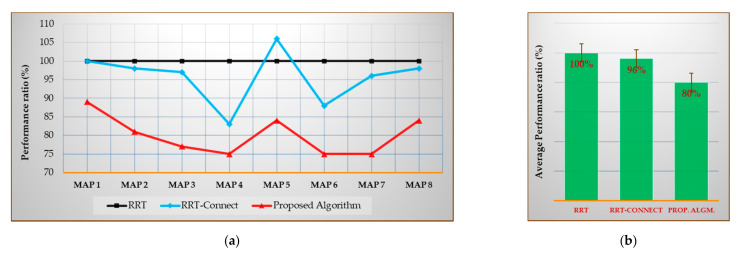
Experimental results in total for the average path length: (**a**) result of each map compared with the RRT algorithm ($x_{cmp}\left(i\right)$); (**b**) average result compared with the RRT algorithm ($X_{cmp}$).

**Figure 21 sensors-21-00333-f021:**
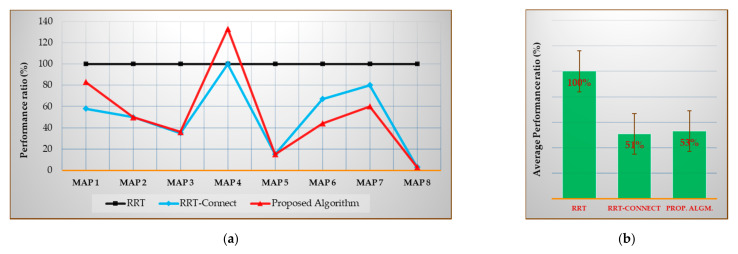
Experimental results in total on the average planning time: (**a**) result of each map compared with the RRT algorithm ($x_{cmp}\left(i\right)$); (**b**) average result compared to the RRT algorithm ($X_{cmp}$).

**Table 1 sensors-21-00333-t001:** Computer performance for simulation.

*H*/*W*	Specification
CPU	Intel Core i7-6700k 4.00 GHz (8 CPUs)
RAM	32,768 MB (32 GB DDR4)
VGA	Nvidia GeForce GTX 1080 (VRAM 8 GB) SLI (x2)

**Table 2 sensors-21-00333-t002:** Experimental result of Map 1 (the parentheses to the right of each value are relative ratios based on RRT 100% ($x_{cmp}\left(1\right)$)).

$\mathrm{Performance}\;(A_{cmp}\left(1\right))$	RRT	RRT-Connect	Proposed Algorithm
Avg. num. of samples (samples)	1216 (100)	729 (60)	823 (68)
Avg. path length (px)	1341 (100)	1343 (100)	1200 (89)
Avg. planning time (ms)	12 (100)	7 (58)	10 (83)

**Table 3 sensors-21-00333-t003:** Experimental result of Map 2 (the parentheses to the right of each value are the relative ratios based on RRT 100% ($x_{cmp}\left(2\right)$)).

$\mathrm{Performance}\;(A_{cmp}\left(2\right))$	RRT	RRT-Connect	Proposed Algorithm
Avg. num. of samples (samples)	271 (100)	101 (37)	113 (42)
Avg. path length (px)	598 (100)	613 (98)	484 (81)
Avg. planning time (ms)	6 (100)	3 (50)	3 (50)

**Table 4 sensors-21-00333-t004:** Experimental result of Map 3 (the parentheses to the right of each value are the relative ratios based on RRT 100% ($x_{cmp}\left(3\right)$)).

$\mathrm{Performance}\;(A_{cmp}\left(3\right))$	RRT	RRT-Connect	Proposed Algorithm
Avg. num. of samples (samples)	6106 (100)	4574 (75)	4679 (77)
Avg. path length (px)	1934 (100)	1871 (97)	1489 (77)
Avg. planning time (ms)	866 (100)	299 (35)	313 (36)

**Table 5 sensors-21-00333-t005:** Experimental result of Map 4 (The parentheses to the right of each value are relative ratios based on RRT 100% ($x_{cmp}\left(4\right)$)).

$\mathrm{Performance}\;(A_{cmp}\left(4\right))$	RRT	RRT-Connect	Proposed Algorithm
Avg. num. of samples (samples)	290 (100)	28 (10)	32 (11)
Avg. path length (px)	711 (100)	588 (83)	534 (75)
Avg. planning time (ms)	3 (100)	3 (100)	4 (133)

**Table 6 sensors-21-00333-t006:** Experimental result of Map 5 (the parentheses to the right of each value are the relative ratios based on RRT 100% ($x_{cmp}\left(5\right)$)).

$\mathrm{Performance}\;(A_{cmp}\left(5\right))$	RRT	RRT-Connect	Proposed Algorithm
Avg. num. of samples (samples)	371 (100)	68 (18)	74 (20)
Avg. path length (px)	554 (100)	588 (106)	465 (84)
Avg. planning time (ms)	13 (100)	2 (15)	2 (15)

**Table 7 sensors-21-00333-t007:** Experimental result of Map 6 (the parentheses to the right of each value are the relative ratios based on RRT 100% ($x_{cmp}\left(6\right)$)).

$\mathrm{Performance}\;(A_{cmp}\left(6\right))$	RRT	RRT-Connect	Proposed Algorithm
Avg. num. of samples (samples)	541 (100)	184 (34)	140 (26)
Avg. path length (px)	886 (100)	778 (88)	668 (75)
Avg. planning time (ms)	9 (100)	6 (67)	4 (44)

**Table 8 sensors-21-00333-t008:** Experimental result of Map 7 (the parentheses to the right of each value are relative ratios based on RRT 100% ($x_{cmp}\left(7\right)$)).

$\mathrm{Performance}\;(A_{cmp}\left(7\right))$	RRT	RRT-Connect	Proposed Algorithm
Avg. num. of samples (samples)	436 (100)	235 (54)	244 (56)
Avg. path length (px)	898 (100)	862 (96)	674 (75)
Avg. planning time (ms)	5 (100)	4 (80)	3 (60)

**Table 9 sensors-21-00333-t009:** Experimental result of Map 8 (The parentheses to the right of each value are the relative ratios based on RRT 100% ($x_{cmp}\left(8\right)$)).

$\mathrm{Performance}\;(A_{cmp}\left(8\right))$	RRT	RRT-Connect	Proposed Algorithm
Avg. num. of samples (samples)	17,033 (100)	3031 (18)	2954 (17)
Avg. path length (px)	1611 (100)	1576 (98)	1358 (84)
Avg. planning time (ms)	4501 (100)	119 (3)	125 (3)

**Table 10 sensors-21-00333-t010:** Experimental results in total for the average number of samples (for first path finding) [%].

Algorithm (*cmp*)	Performance Ratio Based on RRT $\left(x_{cmp}\left(i\right)\right)$								Avg. $\left(X_{cmp}\right)$
	Map 1	Map 2	Map 3	Map 4	Map 5	Map 6	Map 7	Map 8	
RRT	100	100	100	100	100	100	100	100	100
RRT-Connect	60	37	75	10	18	34	54	18	38
Proposed	68	42	77	11	20	26	56	17	40

**Table 11 sensors-21-00333-t011:** Experimental results in total on the average path length (%).

Algorithm (*cmp*)	Performance Ratio Based on RRT $\left(x_{cmp}\left(i\right)\right)$								Avg. $\left(X_{cmp}\right)$
	Map 1	Map 2	Map 3	Map 4	Map 5	Map 6	Map 7	Map 8	
RRT	100	100	100	100	100	100	100	100	100
RRT-Connect	100	98	97	83	106	88	96	98	96
Proposed	89	81	77	75	84	75	75	84	80

**Table 12 sensors-21-00333-t012:** Experimental results in total for the average planning time (%).

Algorithm (*cmp*)	Performance Ratio Based on RRT $\left(x_{cmp}\left(i\right)\right)$								Avg. $\left(X_{cmp}\right)$
	Map 1	Map 2	Map 3	Map 4	Map 5	Map 6	Map 7	Map 8	
RRT	100	100	100	100	100	100	100	100	100
RRT-Connect	58	50	35	100	15	67	80	3	51
Proposed	83	50	36	133	15	44	60	3	53

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
