# Peer review of "Improved RRT-Connect Algorithm Based on Triangular Inequality for Robot Path Planning"

_sensors, 2021, doi:10.3390/s21020333_

Round 1
Reviewer 1 Report
The paper has multiple language issues, which make it difficult to read and understand. It cannot be accepted in its present form. Proofreading is mandatory before submission.
Author Response
Dear Reviewer,
Q1. The paper has multiple language issues, which make it difficult to read and understand. It cannot be accepted in its present form. Proofreading is mandatory before submission.
A1. We apologize for the inconvenience. English quality has been improved through proofreading by a native speaker.

Reviewer 2 Report
It is not clear the main originality since the authors state that "This paper proposes a methodology and an algorithm that applies the Triangular Inequality principle to the RRT-Connect algorithm. The RRT*-Smart and Quick-RRT* algorithms that applied the Triangular Inequality principle to the RRT algorithm are not treated as comparative experiments in this paper".
The significance of the content is high. In this research work, the authors consider the latest research and method used in robot statistical path planning. The quality of presentation is also high. The authors also thoroughly describe the previous algorithm and their pseudo-codes, the impact of each algorithm on the resulting path planning. Finally, they explain how they include the novelty and how it is related to the previous algorithm.
The scientific soundness is very hight. The problem análisis and statement is clear. State of the art and where is their contribution to respect of the related work is also exact, even the method and experimental results.
In summary, I consider that it is an excellent work from a methodological point of view. Still, as I said, I have difficulties in assessing the originality of the work, since the idea of the triangular inequality algorithm seems to be implemented in more advanced RRT algorithms such as RRT*, and this place this work in a weak position. I suggest the authors reinforce this point. For example, is there any advantage of RRT comparing with RRT*? Does it make sense to investigating in RRT algorithm?
Author Response
Dear Reviewer,
Q1. In summary, I consider that it is an excellent work from a methodological point of view. Still, as I said, I have difficulties in assessing the originality of the work, since the idea of the triangular inequality algorithm seems to be implemented in more advanced RRT algorithms such as RRT*, and this place this work in a weak position. I suggest the authors reinforce this point. For example, is there any advantage of RRT comparing with RRT*? Does it make sense to investigating in RRT algorithm?
A1. The RRT* algorithm is merely a derivative of the RRT algorithm. The RRT* algorithm has the Rewiring (Search for the parent node where path distance of qnew to qstart is most optimized and change the neighboring nodes to optimize the path distance) and the Neighbor search (Search for nearby nodes of qnew) processes to obtain a shorter path length than the RRT algorithm [A]. However, there is a trade-off in efficiency in this process. In other words, the optimality (shorter path length) has improved, but the Convergence time (also, planning time) is significantly increased [B]. Therefore, the RRT* algorithm cannot be said to be better than the RRT algorithm in all performance measures, and it can be said to be the RRT* algorithm improves optimality at the expense of planning time. As a result, this paper shows that the proposed algorithm improves the optimality of the RRT and RRT-Connect algorithms without sacrificing other performance measures such as Number of Samples and planning time.
[A] Karaman, Sertac, and Emilio Frazzoli. "Sampling-based algorithms for optimal motion planning." The international journal of robotics research 30.7 (2011): 846-894.
[B] Noreen, Iram, Amna Khan, and Zulfiqar Habib. "A comparison of RRT, RRT* and RRT*-smart path planning algorithms." International Journal of Computer Science and Network Security (IJCSNS) 16.10 (2016): 20.
The above content has been added to the Introduction and we have made changes to make it easier for readers to understand generally: pp. 2, Line: 55 and 74.

Reviewer 3 Report
The paper proposes the use of triangle inequality to reduce the path length obtained through RRT-connect method. The idea of the paper is interesting. However, the following issues need to be solved:
- The paper is too long, having the length of a book chapter and not of a journal paper. I suggest to shrink some parts of the paper that lay outside the goals of the paper. An example: why the authors present the basic RRT method in details (RRT is a well-known method, published 22 years ago) since it is used only for comparison purposes? A reference and the basic idea of RRT are sufficient. The same happens with RRT-Connect: instead of focusing on the differences between RRT-connect and the proposed method, the paper makes an exhaustive presentation of RRT-connect (also a well-known method, published 20 years ago).
- English needs serious polishing. There are numerous grammar mistakes, typos and badly-shaped sentences. Let’s have a look only to the abstract: a) the first sentence is badly-shaped, instead of “proposes” it is written “proposed”, the words “Triangular” and “Inequality” are capitalized without a reason, the sequence “guarantees the convergence time than the RRT algorithm” is confusing; b) the second sentence: the sequence “this paper compared with the RRT and RRT-Connect algorithms in various environments through simulation” is strange; and so on. Please check the entire manuscript.
- The title of the paper includes the sequence "Enhancement of Optimality" which is somehow curious. "Optimality" cannot be "enhanced". "Optimal" is non-gradable so we cannot say that A is more optimal than B, Moreover, no optimization criteria is presented to understand what the authors mean by this. In addition, throughout the manuscript, there are many formulations that contravene the fact that “Optimal” is non-gradable. See for example in line 16: “better optimality”, “more optimal”; line 224: “increasing the optimality”; line 225: “improve their optimality”, line 230: “more optimal” and many-many more.
- what a percentage over 100% in Figures 19b and 20b (the limits of the green bar (marked by the small vertical brown line) for RRT) means? Moreover, Y-axis is not specified in Fig. 19b and 20b.
- Some details about convergence of the proposed algorithm are needed;
- The sequence “Total experimental results” in the capture of Tables 10-12 or figures 19-20 is confusing and needs to be replaced;
- The meaning of “completeness” when speaking about robot path needs to be explained.
Author Response
Dear Reviewer,
Q1. The paper is too long, having the length of a book chapter and not of a journal paper. I suggest to shrink some parts of the paper that lay outside the goals of the paper. An example: why the authors present the basic RRT method in details (RRT is a well-known method, published 22 years ago) since it is used only for comparison purposes? A reference and the basic idea of RRT are sufficient. The same happens with RRT Connect: instead of focusing on the differences between RRT-connect and the proposed method, the paper makes an exhaustive presentation of RRT-connect (also a well known method, published 20 years ago).
A1. We also had concerns about this, but for two reasons, we wrote the RRT and RRT-Connect algorithms in detail. First, as we did a performance comparison experiment through simulation rather than an actual robot experiment, we needed to write in-detail the Pseudocode used in our unverified simulator. Second, since the triangular inequality method proposed in this paper is applied inside the RRT-Connect algorithm, it was necessary to deal with the RRT-Connect algorithm in detail, and the RRT algorithm is helpful in understanding the pseudocode of the RRT-Connect algorithm written in this paper. However, reflecting the opinions of the reviewer, the pseudocode of the RRT algorithm (originally 2.1), the description of the internal functions (originally 2.2, 2.3) used in their pseudocodes, and basic mathematical modeling of the RRT algorithm (originally 4.2.1), which are not necessary to explain the RRT-Connect and the proposed algorithm, have been moved to the Appendix (currently, A.1–A.3).
Q2. English needs serious polishing. There are numerous grammar mistakes, typos and badly-shaped sentences. Let’s have a look only to the abstract: a) the first sentence is badly-shaped, instead of “proposes” it is written “proposed”, the words “Triangular” and “Inequality” are capitalized without a reason, the sequence “guarantees the convergence time than the RRT algorithm” is confusing; b) the second sentence: the sequence “this paper compared with the RRT and RRT-Connect algorithms in various environments through simulation” is strange; and so on. Please check the entire manuscript.
A2. English quality has been improved through proofreading by a native speaker. The expressions pointed out have also been corrected as appropriate.
Q3. The title of the paper includes the sequence "Enhancement of Optimality" which is somehow curious. "Optimality" cannot be "enhanced". "Optimal" is non-gradable so we cannot say that A is more optimal than B, Moreover, no optimization criteria is presented to understand what the authors mean by this. In addition, throughout the manuscript, there are many formulations that contravene the fact that “Optimal” is non-gradable. See for example in line 16: “better optimality”, “more optimal”; line 224: “increasing the optimality”; line 225: “improve their optimality”, line 230: “more optimal” and many-many more.
A3. The meaning of ‘Optimality’ as we intended it to be used is as a measure of how close a path length to optimal. However, as the Reviewer pointed out, it seems that the meaning was misunderstood and written incorrectly. Including the paper title, ‘Optimality’ and ‘Optimal’, which were used incorrectly, have been changed to appropriate expressions such as ‘Shorter path length’ along with overall English correction.
Q4. what a percentage over 100% in Figures 19b and 20b (the limits of the green bar (marked by the small vertical brown line) for RRT) means? Moreover, Y-axis is not specified in Fig. 19b and 20b.
A4. Figure 19 (a), Figure 20 (a), and Figure 21 (a) are diagrams showing the performance of the RRT-Connect algorithm and the proposed algorithm when the performance of the RRT algorithm is set as 100% for each environment map. In (b) of each figure shows the performance average of all environment maps for each algorithm. The values shown in (a) of Figure 19-21 can be expressed as in Equation A-B, and the values shown in (b) can be expressed as Equation C as below:
Acmp(i) = ∑Tk=0(acmp_k(i))/T, (A)
Here, refers to the performance value of each algorithm shown in Table 2-9, cmp is the algorithm to be calculated, i is the index of the environment map (X-axis in Figure 19-21 (b)), k is the repeat index, and T is the number of repeats ( is the value of the performance measure a when the k-th implementation of the cmp algorithm in Map i). In the experiment in this paper, 50 repetitions are performed. That is, Equation A shows the average value of the performance when it is repeated T times to check the performance of a certain algorithm in Map i,
xcmp(i) = Acmp(i)/ARRT(i), (B)
where refers to the Y-axis in Figure 19-21 (a), and A is the value for the corresponding performance measure of the algorithm to be compared ( is the value of the RRT algorithm),
Xcmp = ∑Mi=0(x(i)cmp)/M, (C)
where refers to the Y-axis in Figure 19-21 (b), and M is the number of environment maps used in the experiment. In the experiment in this paper, eight maps are used. That is, Equation C shows the average value of i for all maps in Equation B. The above has been added to the first part of Section 5.2 (pp. 19, Line: 505) and the first part of Section 5.3 (pp. 25, Line: 645). Also, each Y-axis has been added in (b) of Figure 19-21.
Q5. Some details about convergence of the proposed algorithm are needed.
A5. The meaning of ‘Convergence time’ as we used to be how quickly the path can be planned from the start point to the goal point. However, we see that this was a misuse of the term. The scope of research we are going to cover is how much faster and shorter to find the path. This is because, in a dynamic environment, it is more important to find a path that can be navigated. In a dynamic environment, there may not be enough time until convergence. In other words, the purpose of our proposed algorithm is to improve the RRT-Connect algorithm so that it can find a shorter path during the same planning time (computation time for first path finding). Incorrectly written expressions were replaced, such as ‘Convergence time’ replaced with ‘Planning time’, and an explanation of the ‘Convergence rate’ was added to the introduction (pp. 1, Line: 42), and explanation of why we use ‘Planning time’ as a measure of performance was written in detail: pp. 2, Line: 77.
Q6. The sequence “Total experimental results” in the capture of Tables 10-12 or figures 19-20 is confusing and needs to be replaced.
A6. Tables 10–12 are the data tables for Figures 19-21. The explanations for these are included in Answer A4 to Q4 above, and each of these tables and figures have captions marked with symbols so that reader can see the value more clearly.
Q7. The meaning of “completeness” when speaking about robot path needs to be explained.
A7. In the introduction, the definition of ‘Completeness’ was briefly explained. If completeness is not guaranteed by the robot path planning algorithm, there is a problem that the path cannot be found during a finite amount of time. This is a very fatal problem in the robot path planning. The above has been added to the introduction: pp. 1, Line: 31.

Round 2
Reviewer 1 Report
The language has been improved, but still, there are some issues.
No comments on the paper content - the results look very good. However, I would suggest adding a critical review: what are the assumptions and limitations of the suggested algorithm?
It is still necessary to do proofreading by a native speaker. Some sentences are kind of strange.
For example:
"The last is 27 the Artificial Potential Field algorithm [5], which creates an artificial potential field and moves the robot to the goal according to the flow of potential power." The algorithm doesn't move the robot (maybe "directs"?).
"These classical algorithms include Optimality, which means always ensuring the optimal path, Clearance, which indicates a lower probability of collision between obstacles and a robot, and 31 Completeness, which means that if a path exists it can always be found; these three are considered important and have been the main focus of study [6]." An algorithm cannot include optimality, etc. These are criteria or constraints, that are taken into account or guaranteed (as you correctly say in the next sentence) by algorithms.
46: an optimal path -> the optimal path
Figure 1(c) yellow-brown text ("Rewiring") is unreadable.
Author Response
Dear Reviewer,
Q1. I would suggest adding a critical review: what are the assumptions and limitations of the suggested algorithm?
A1. In response to the opinions of the reviewer, the assumptions of the proposed algorithm have been added were Line 208, pp. 8:
“The proposed triangular inequality-based RRT-Connect algorithm requires the following assumptions.
[Assumptions]
- There is only one start point and one goal point even though the goal point may be changed incrementally as time goes on.
- The robot is capable of omnidirectional motion.”
Also, the limitation of the proposed algorithm has been added were Line 723, pp. 29:
“However, one of the limitations of the proposed algorithm is the Kinodynamic planning problem [17]. When the intermediate node disappears by Triangular-Rewiring method, a non-differentiable piecewise linear section with sharp corner may occurs, which cause a problem related with the kinematic constraint of the robot.”
Q2. "The last is the Artificial Potential Field algorithm [5], which creates an artificial potential field and moves the robot to the goal according to the flow of potential power." The algorithm doesn't move the robot (maybe "directs"?).
A2. As following the reviewer’s comment, the sentence "The last is the Artificial Potential Field algorithm [5], which creates an artificial potential field and moves the robot to the goal according to the flow of potential power." has been changed into "The last is the Artificial Potential Field algorithm [5], which creates an artificial potential field and directs the robot to the goal according to the flow of potential power." (Line 27, pp. 1)
Q3. "These classical algorithms include Optimality, which means always ensuring the optimal path, Clearance, which indicates a lower probability of collision between obstacles and a robot, and Completeness, which means that if a path exists it can always be found; these three are considered important and have been the main focus of study [6]." An algorithm cannot include optimality, etc. These are criteria or constraints, that are taken into account or guaranteed (as you correctly say in the next sentence) by algorithms.
A3. As following the reviewer’s comment, the sentence "These classical algorithms include Optimality, which means always ensuring the optimal path, Clearance, which indicates a lower probability of collision between obstacles and a robot, and Completeness, which means that if a path exists it can always be found; these three are considered important and have been the main focus of study [6]." has been changed into “Optimality means always ensuring the optimal path. Clearance indicates a lower probability of collision between obstacles and the robot. Completeness means that if a path exists, it can always be found. Optimality, Clearance and Completeness are considered important in these classical algorithms and have been the main focus of study [6].” (Line 29, pp. 1)
Q4. 46: an optimal path -> the optimal path
A4. As following the reviewer’s comment, the sentence “…an optimal path.” has been changed into “…the optimal path.” (Line 46, pp. 2)
Q5. Figure 1(c) yellow-brown text ("Rewiring") is unreadable.
A5. The text (“triangularRewiring”) color has been darkened to make it more readable in Fig. 1(c). In addition, Fig. 5(b), Fig. 7, Fig. 8, Fig. 9(c), and Fig. 9(d) have also been changed, similarly.

Reviewer 3 Report
The manuscript has been significantly improved. The authors have successfully solved all my previous comments/concerns. However, some minor issues need to be solved:
- lines 33-34: the sequence "there is a problem that the path cannot be found even if the path is planned for a finite amount of time" is confusing and needs to be reshaped.
- line 67: The word "ancestral" is not appropriate to the context;
- line 149: The word "currently" is also confusing in the context;
- line 446: Please check if the comma between "average" and "Number" is needed;
Author Response
Dear Reviewer,
Q1. lines 33-34: the sequence "there is a problem that the path cannot be found even if the path is planned for a finite amount of time" is confusing and needs to be reshaped.
A1. As following the reviewer’s comment, the sentence "there is a problem that the path cannot be found even if the path is planned for a finite amount of time" has been changed into “there is a problem that the path may not be found in finite time.” (Line 33, pp. 1)
Q2. line 67: The word "ancestral" is not appropriate to the context;
A2. As following the reviewer’s comment, the sentence “this paper proposes a triangular inequality-based RRT-Connect algorithm that finds and wires the highest ancestral existing node that can be wired while alternately expanding in two trees rooted at the starting and goal points.” has been changed into “this paper proposes a triangular inequality-based RRT-Connect algorithm that finds an ancestor node as a via point, where the addition of path length from the start point to the via point and path length from the via point to the newly inserted node is the most optimized, based on the principle of triangular inequality and RRT-Connect.” (Line 69, pp. 2)
Q3. line 149: The word "currently" is also confusing in the context;
A3. As following the reviewer’s comment, the sentence “When a path is created in the Connect method, the distance dreach is calculated for the path Preach to qstart and qgoal. Currently, if dreach is smaller than the path length dshorter or reached first (dshorter = 0), the resultant path R becomes Preach, and dshorter becomes dreach.” has been changed into “When a path is created by the Connect method, the distance dreach is calculated for the path Preach from qstart to qgoal. At this time, if dreach is smaller than dshorter(the shortest path length until now) or Preach is the first path found (i.e., dshorter = 0), the resultant path R becomes Preach, and dshorter becomes dreach.” (Line 150, pp. 5)
Q4. line 446: Please check if the comma between "average" and "Number" is needed;
A4. As following the reviewer’s comment, the sentence, “…and the compared performance measure is the average, Number of sampling (samples), Path length (pixels), and Planning time (milliseconds) for all trials for 50 times from the start point to the goal position until the first path has been found).” has been changed into “…and the performance measures used for comparison of various algorithms are Number of sampling (samples), Path length (pixels), and Planning time (milliseconds). And each performance measure is experimented with 50 trials from the same start point to the same goal point until the first path has been found).” (Line 459, pp. 18)
